# PREDICTIVE INVERSE DYNAMICS MODELS ARE SCALABLE LEARNERS FOR ROBOTIC MANIPULATION

**Yang Tian**[1,4,*]   **Sizhe Yang**[6,1,*]   **Jia Zeng**[1]   **Ping Wang**[3,4,5]   **Dahua Lin**[6]
**Hao Dong**[2]   **Jiangmiao Pang**[1,†]

[1] Shanghai AI Laboratory    [2] CFCS, School of CS, Peking University
[3] National Engineering Research Center for Software Engineering, Peking University
[4] School of Software & Microelectronics, Peking University
[5] Key Laboratory of High Confidence Software Technologies (PKU), Ministry of Education
[6] Chinese University of Hong Kong
[*] Equal contribution, Author ordering determined by coin flip    [†] Corresponding authors
Project page: https://nimolty.github.io/Seer/

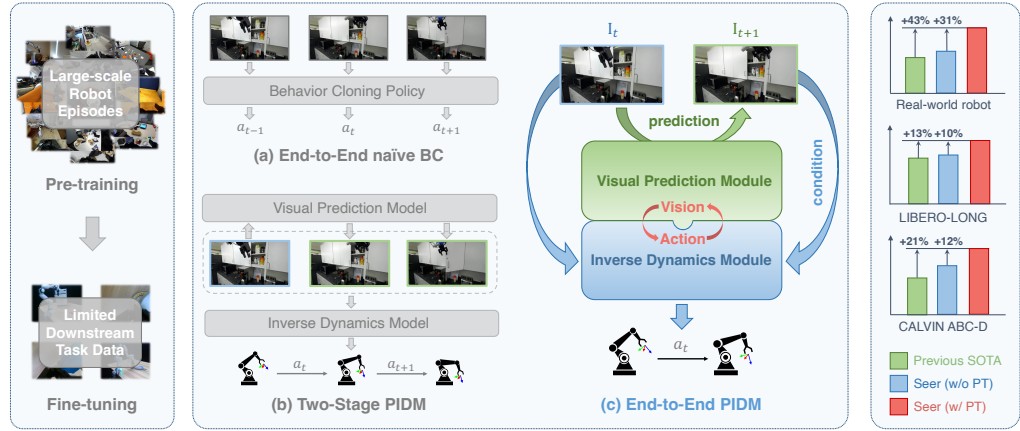

Figure 1: In contrast to previous methods that (a) conduct end-to-end naive behavior cloning from large-scale robotic data or (b) use decoupled visual prediction and inverse dynamics models to set goals and guide actions, we present end-to-end Predictive Inverse Dynamics Models (PIDM) that closes the loop between vision and action. Seer, the model we built, surpasses previous states of the art and demonstrates consistent improvements over the ablated version without pre-training.

## ABSTRACT

Current efforts to learn scalable policies in robotic manipulation primarily fall into two categories: one focuses on "action," which involves behavior cloning from extensive collections of robotic data, while the other emphasizes "vision," enhancing model generalization by pre-training representations or generative models, also referred to as world models, using large-scale visual datasets. This paper presents an end-to-end paradigm that predicts actions using inverse dynamics models conditioned on the robot's forecasted visual states, named Predictive Inverse Dynamics Models (PIDM). By closing the loop between vision and action, the end-to-end PIDM can be a better scalable action learner. In practice, we use Transformers to process both visual states and actions, naming the model Seer. It is initially pre-trained on large-scale robotic datasets, such as DROID, and can be adapted to real-world scenarios with a little fine-tuning data. Thanks to large-scale, end-to-end training and the synergy between vision and action, Seer significantly outperforms previous methods across both simulation and real-world experiments. It achieves improvements of 13% on the LIBERO-LONG benchmark, 21% on CALVIN ABC-D, and 43% in real-world tasks. Notably, Seer sets a new state-of-the-art on CALVIN ABC-D benchmark, achieving an average length of 4.28, and exhibits superior generalization for novel objects, lighting conditions, and environments under high-intensity disturbances on real-world scenarios. Code and models are publicly available at https://github.com/OpenRobotLab/Seer/.

# 1 INTRODUCTION

Learning scalable and generalizable policies has become a main focus in robotic manipulation. Recent efforts primarily fall into two categories: one focuses on "action," like RT-1 (Brohan et al., 2022), Octo (Ghosh et al., 2024), and OpenVLA (Kim et al., 2024), which perform naive behavior cloning from large-scale robotic data such as Open X-Embodiment and DROID (Padalkar et al., 2024; Khazatsky et al., 2024). The other emphasizes "vision" and may learn representations through discriminative or generative ways and integrate with the control policy in a two-stage manner. For example, R3M (Nair et al., 2022) and MVP (Xiao et al., 2022) learn discriminative representations from large-scale video datasets such as Ego4D (Grauman et al., 2022), while UniPI (Du et al., 2024), Susie (Black et al., 2023), and CLOVER (Bu et al., 2024) develop generative models as "world models" to facilitate manipulation policies. Apparently, the scaling laws in robot learning are still evolving, with researchers exploring strategies through diverse data and methods.

We revisit these approaches and propose that a scalable manipulation policy should integrate vision and action in a closed loop. This integration is natural and necessary, as humans typically coordinate their hands and eyes to manipulate objects. Therefore, closing the loop during training and inference are both necessary for a better scalable action learner.

This paper achieves this by introducing a simple yet effective end-to-end Predictive Inverse Dynamics Models (PIDM) that can unify the advantages of previous methods. As shown in Figure 1, it predicts actions using Inverse Dynamics Models (IDM) conditioned on the "Predictive" visual states of the robot. During training, both the visual prediction module and the inverse dynamics module are optimized synergistically in an end-to-end manner. During inference, PIDM ensures continuous synergy between vision and action at each execution step. In contrast to previous methods that use IDM, our approach is the first to optimize vision and action in an end-to-end manner. Throughout this paper, unless otherwise specified, PIDM is assumed to be end-to-end.

In practice, we use Transformers to process both visual states and actions and name the model Seer. Seer benefits from PIDM by simultaneously leveraging visual, temporal, and action information from large-scale datasets, and can be more scalable due to the Transformer architecture. We introduce a foresight token to predict future RGB images and an action token to estimate intermediate actions between current and predicted future observations. Both tokens are fused with input RGB images, robot state, and language tokens through a multi-modal encoder. Importantly, we design a unidirectional attention mask that allows the action token to deeply integrate past and future predictive information, facilitating end-to-end training.

We conduct extensive experiments on both simulation and real-world benchmarks. On two widely adopted simulation benchmarks, LIBERO-LONG (Liu et al., 2024) (10 tasks) and CALVIN ABC-D (Mees et al., 2022) (34 tasks), our method demonstrates a 10.4% improvement in success rate and a 0.75 increase in average task completion length compared to state-of-the-art baselines. Our results further indicate superiority in long-horizon task completion, unseen scene generalization, and data efficiency. Additionally, We evaluate our method on six challenging real-world tasks with over 900 trials. Leveraging the public large robot dataset DROID (Khazatsky et al., 2024), our method consistently shows robustness, even under disturbances and with limited fine-tuning data.

# 2 RELATED WORK

**Action-Centric Pre-training for Manipulation.** Recent advancements in action-centric pre-training have significantly enhanced manipulation policies. Approaches like SMART (Sun et al., 2023) and DualMind (Wei et al., 2023) emphasize understanding the dynamics within environments. Some studies (Agrawal et al., 2016; Brandfonbrener et al., 2024) integrate current and goal information to extract effective features or serve as an auxiliary objective. Subsequently, a standard behavior cloning approach is applied during downstream task implementations. Additionally, RT-X (Padalkar et al., 2024) and Octo (Ghosh et al., 2024) focus on pre-training robot policies using diverse datasets to facilitate extensive generalization capabilities. Recently, vision-language models (VLMs) have demonstrated considerable common-sense knowledge about the world and strong capabilities in understanding both language and images. OpenVLA (Kim et al., 2024) further pre-trains VLMs using robotic data, leveraging their prior knowledge to achieve robust performance on downstream language-conditioned manipulation tasks. While these methods primarily supervise

actions, they do not fully exploit the rich visual and temporal information inherent in robot demonstrations. In contrast, we pre-train policies by integrating conditional visual foresight and inverse dynamics prediction, allowing for comprehensive utilization of robotic data.

**Vision-Centric Pre-training for Manipulation.** Extensive research has focused on visual pre-training for visuomotor control (Karamcheti et al., 2023; Zeng et al., 2024). One major direction involves representation learning using techniques such as masked image modeling (Xiao et al., 2022; Radosavovic et al., 2023; Seo et al., 2023), contrastive learning (Nair et al., 2022; Ma et al., 2022; 2023), and generative video pre-training (Wu et al., 2024). Another line of work focuses on visual expectations guiding actions, termed the Predictive Inverse Dynamics Model (PIDM) (Bharadhwaj et al., 2024; Wang et al., 2024; Soni et al., 2024; Chen et al., 2024). Firstly, a video generation model predicts future visual sub-goals and is pre-trained on general visual datasets. Then, an inverse dynamics (goal-conditioned) low-level policy is trained on downstream robot data to predict the intermediary actions. Compared to these two-stage PIDM, we propose an end-to-end PIDM paradigm that leverages large-scale robot data for pre-training, showing better performance.

**Pre-training Datasets for Manipulation.** High-quality, large-scale, and diverse pre-training data is crucial for acquiring manipulation skills. Image datasets (Deng et al., 2009), video datasets (Damen et al., 2018; Goyal et al., 2017); (Grauman et al., 2022), and robot datasets are commonly utilized for this purpose. Image datasets provide rich semantic information, while video datasets contain temporal information. Both enhance visual representations for manipulation, however, their lack of action labels and robot states limits their utility in decision-making. Some studies focus on collecting robot behavior data (Mandlekar et al., 2019; Walke et al., 2023; Dasari et al., 2019; Bahl et al., 2023; Jang et al., 2022), but the data diversity remains relatively constrained. Recent efforts aim to further scale and diversify robot datasets. For instance, the Open X-Embodiment dataset (Padalkar et al., 2024) aggregates data from 22 different robots across 21 institutions, covering 527 skills and 160,266 tasks. DROID (Khazatsky et al., 2024) includes 76,000 trajectories collected across 564 scenes and 86 tasks. In this work, we leverage DROID to pre-train policies for real-world validation, demonstrating that rich behavioral data significantly enhances success rates in downstream tasks.

## 3 METHOD

In this section, we describe Seer in detail. We begin with a brief problem formulation (Section 3.1). Next, we discuss keys in our end-to-end PIDM—conditional visual foresight and inverse dynamics prediction (Section 3.2), enabling Seer to forecast the future and adjust actions accordingly. We then elaborate on the carefully designed model architecture (Section 3.3), through which we formulate Seer in an end-to-end manner. Finally, we provide implementation details (Section 3.4).

### 3.1 PROBLEM FORMULATION

Given a large-scale dataset of diverse manipulation demonstrations $D_1 = \{(l, o_t, s_t, a_t)_{t=0}^{T_i}\}_{i=0}^{N_1}$, and a smaller downstream dataset $D_2 = \{(l, o_t, s_t, a_t)_{t=0}^{T_j}\}_{j=0}^{N_2}$ (where $N_1 >> N_2$), our goal is to enhance downstream task performance through effective pre-training on $D_1$, followed by fine-tuning on $D_2$. Each trajectory $\{(l, o_t, s_t, a_t)_{t=0}^{T}\}$ provides the time step $t$, language instruction $l$, RGB images $o_t$ from the eye-on-hand and eye-on-base views, robot states $s_t$ and robot actions $a_t$, which include arm actions $a_{\text{arm}}$ (6D pose) and gripper actions $a_{\text{gripper}}$ (open or close). It is important to note that current large pre-training robot data may contain incomplete language annotations $l$ and task-agnostic actions $a_t$, such as random exploration in the environment (Mees et al., 2022). However, Seer could handle this scenario effectively due to the following specific design choices.

### 3.2 END-TO-END PIDM

**Vision: Conditional Visual Foresight.** A key insight is that informative future states guide actions. Therefore, we propose conditional visual foresight $f_{\text{fore}}$ to effectively anticipate future visual representations. Seer takes as input a goal $g$ in the form of language instructions or robot states, along with historical observations $h_t$, and predicts the RGB images at the time step $t + n$, denoted by $\hat{o}_{t+n}$

$$\hat{o}_{t+n} = f_{\text{fore}}(g, h_t). \tag{1}$$

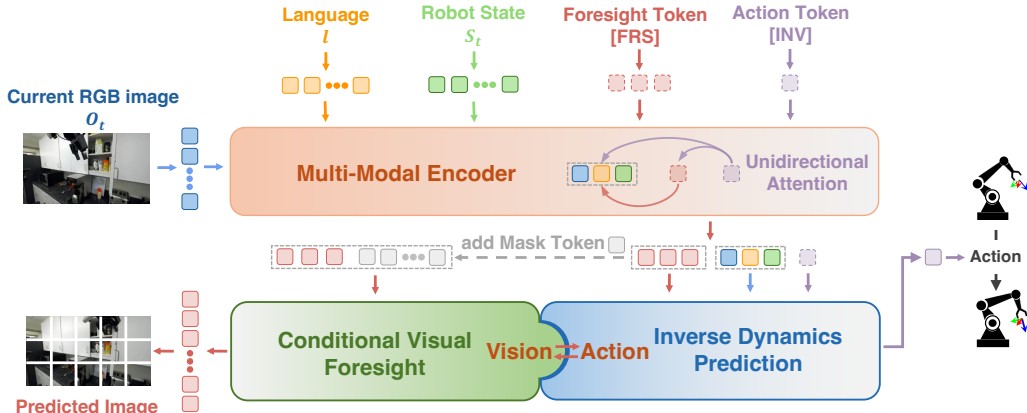

Figure 2: **Pipeline of Seer.** Seer consists of three parts: Multi-Modal Encoder, Conditional Visual Foresight and Inverse Dynamics Prediction. In Multi-Modal Encoder, Seer incorporates the foresight token [FRS] and the action token [INV]. Both tokens attend to the RGB images, language tokens, and robot state tokens, with [INV] also attending to [FRS]. In Conditional Visual Foresight, the encoded [FRS], along with new mask tokens, aims to reconstruct future RGB images. In Inverse Dynamics Prediction, the encoded [INV] and other tokens speculate intermediary actions.

The historical observations $h_t$ consist of the RGB images $o_{t-m+1:t}$ and robot states $s_{t-m+1:t}$ over the last $m$ time steps. Due to the rich information contained in RGB images, their abundance, and easy accessibility, we select them as future representatives. Following (He et al., 2022), the loss function $\mathcal{L}_{\text{fore}}$ computes the mean squared error (MSE) at the pixel level

$$\mathcal{L}_{\text{fore}} = \|f_{\text{fore}}(g, h_t) - o_{t+n}\|_2^2. \tag{2}$$

**Action: Inverse Dynamics Prediction.** Given two temporally ordered observations $o_t$ and $o_{t+1}$, inverse dynamics prediction estimates the intermediate action $\hat{a}_t$. Here, we extend inverse dynamics $f_{\text{inv}}$ to predict the action sequence $\hat{a}_{t:t+n-1}$ given goal $g$, historical observations $h_t$ and $o_{t+n}$. Specifically, we replace ground truth $o_{t+n}$ with the predicted representation $\hat{o}_{t+n}^l$ in the latent space

$$\hat{a}_{t:t+n-1} = f_{\text{inv}}(g, h_t, \hat{o}_{t+n}^l). \tag{3}$$

The loss function $\mathcal{L}_{\text{inv}}$ comprises the arm action loss $\mathcal{L}_{\text{arm}}$ and the gripper action loss $\mathcal{L}_{\text{gripper}}$

$$\mathcal{L}_{\text{inv}} = \mathcal{L}_{\text{arm}} + \lambda \mathcal{L}_{\text{gripper}}, \tag{4}$$

where $\mathcal{L}_{\text{arm}}$ is a Smooth-L1 loss, $\mathcal{L}_{\text{gripper}}$ is a Binary Cross Entropy (BCE) loss and $\lambda$ is set to 0.01.

**Close the Loop between Vision and Action.** Seer integrates conditional visual foresight with inverse dynamics prediction effectively through training, enabling full utilization of both vision and action information in robot data. In detail, $f_{\text{fore}}$ incorporates a clear goal $g$ and historical observations $h_t$ to predict future RGB images $\hat{o}_{t+n}$. A latent representation $\hat{o}_{t+n}^l$ (leading to $\hat{o}_{t+n}$) and $h_t$ facilitate action prediction via $f_{\text{inv}}$. Due to the model design of Seer, all these processes are executed in an end-to-end manner. The overall training loss $\mathcal{L}$ comprises $\mathcal{L}_{\text{fore}}$ and $\mathcal{L}_{\text{inv}}$

$$\mathcal{L} = \alpha \mathcal{L}_{\text{fore}} + \mathcal{L}_{\text{inv}}, \tag{5}$$

where $\alpha$ is a hyperparameter set to 0.5. Compared to single-step action prediction, predicting multiple steps provides temporal action consistency and robustness to idle actions (Chi et al., 2023). During inference, we can either discard actions beyond the first step or apply temporal ensemble techniques to compute a weighted average of the multi-step actions.

### 3.3 MODEL ARCHITECTURE

**Input Tokenizers.** As illustrated in Figure 2, the model processes three types of inputs: language, images, and robot states. We use different encoders to tokenize each modality accordingly. For language inputs, we first tokenize the text and then use a CLIP text encoder (Radford et al., 2021) to obtain text embeddings, which are subsequently projected into a latent space using a linear layer. For

image inputs, the images are first patchified and passed through a pre-trained Vision Transformer (ViT) (He et al., 2022) to generate visual embeddings. Since the ViT produces hundreds of embeddings per image, imposing a significant computational burden on the transformer backbone, and much of the visual information is irrelevant to the manipulation task, we employ a perceiver resampler (Alayrac et al., 2022) to extract task-relevant visual features and reduce the number of image tokens. For the robot state, we encode it into state tokens using a multi-layer perceptron (MLP).

**Multi-Modal Encoder.** The multi-modal encoder in our model is based on a GPT-2 style transformer architecture. Before feeding the sequential language-image-state pairs into the transformer, we append readout tokens [INV] and [FRS] to each timestep. These readout tokens attend to embeddings from different modalities, serving as image and action latents used for conditional visual foresight and inverse dynamics prediction. To incorporate temporal information, we also add a learnable position embedding to the tokens for each timestep.

The [FRS] token aims to facilitate conditional visual foresight, corresponding to aforementioned $\hat{o}_{t+n}^l$. It attends to language, historical image and state tokens. Conversely, the [INV] token performs inverse dynamics prediction conditioned on the predicted visual foresight, attending to the input tokens and, crucially, the foresight token [FRS]. This special unidirectional attention mask in a transformer encoder, highlighted in Figure 2, brings two benefits. First, this will help the [INV] token deeply integrate both past and future predictive information within a multi-layer network. Second, this enables an end-to-end training paradigm through the fusion in the latent space.

**Readout Decoders.** Encoded by the multi-modal encoder, the action and image latents generated by the [INV] and [FRS] readout tokens are fed into the readout decoders to predict images and actions. The action decoder utilizes an MLP to transform the action latent into the action vector $a_t$. For image decoding, we employ a vision transformer (ViT) as the image decoder, following (He et al., 2022). The image decoder takes the image latent along with masked tokens as input. Processed by ViT, the output corresponding to each masked token represents a specific patch of the image.

### 3.4 IMPLEMENTATION DETAILS

**Training.** The training objectives, conditional visual foresight and inverse dynamics prediction, remain consistent between pre-training and fine-tuning. Notably, two key differences in model configurations exist between these phases. First, missing language instructions are common in robotic pre-training datasets. In such cases, during pre-training, the robot state token at the future time step $t + n + 1$ acts as a goal. The [FRS] would attend to it instead of the language token, ensuring [FRS] to acquire unambiguous information. Second, pre-training data may include random or meaningless behaviors, such as environmental exploration. Consequently the [INV] and [FRS] tokens do not attend to previous image and robot state tokens to prevent overfitting to any specific behaviors.

**Inference.** During inference, the complete language instruction $l$, robot states $s$, and image observations $o$ are provided as inputs. The [FRS] token attends to the historical image, state, and language instruction tokens to perform conditional visual foresight, predicting the future images. In turn, the [INV] token attends to the input tokens and one more foresight [FRS] token to perform inverse dynamics prediction, outputting the action. Further details are available in the Appendix.

**Model.** Throughout the training process, the pre-trained visual and text encoders are kept frozen, comprising a total of 251M untrainable parameters. The rest components are fully trainable. The standard version of Seer possesses 65M trainable parameters. Additionally, we have scaled up the parameter size and developed a Seer-Large variant, which contains 315M trainable parameters. Unless specified otherwise, the mention of Seer refers to the version with 65M trainable parameters.

## 4 SIMULATION EXPERIMENTS

We conduct experiments on two simulation benchmarks LIBERO-LONG (Liu et al., 2024), CALVIN ABC-D (Mees et al., 2022). Our aim is to answer: 1) How does our method perform on challenging simulation benchmarks? 2) Does our pipeline maintain consistent effectiveness as the amount of downstream fine-tuning data varies? 3) Are the training objectives in Seer effective?

Table 1: **LIBERO-LONG results.** For each task, we present the average performance of top-3 checkpoints averaged over 20 rollouts. The metric "Avg. Success" measures the average success rate across ten tasks. Seer outperforms baselines with higher Avg. Success and better results on most tasks. The best results are **bolded**.

| Method | Avg. Success ↑ | Put soup and box in basket | Put box and butter in basket | Turn on stove and put pot | Put bowl in drawer and close it |
|---|---|---|---|---|---|
| MTACT | 41.0 | 30.0 | 50.0 | 75.0 | 85.0 |
| MVP | 68.2 | 83.3 | 90.0 | 80.0 | 88.3 |
| MPI | 77.3 | 66.6 | 86.6 | 96.6 | 95.0 |
| OpenVLA | 54.0 | 35.0 | **95.0** | 65.0 | 45.0 |
| Seer (scratch) | 78.7 | 80.0 | 90.0 | 91.7 | 81.7 |
| Seer | **87.7** | **91.7** | 90.0 | **98.3** | **100** |

| Put mugs on left and right plates | Pick book and place it in back | Put mug on plate and put pudding to right | Put soup and sauce in basket | Put both pots on stove | Put mug in microwave and close it |
|---|---|---|---|---|---|
| 20.0 | 75.0 | 0.00 | 0.00 | 10.0 | 65.0 |
| 46.7 | 63.3 | 45.0 | 78.3 | 60.0 | 46.7 |
| 83.3 | 83.3 | 56.6 | 86.6 | 40.0 | **78.3** |
| 40.0 | 80.0 | 60.0 | 45.0 | 20.0 | 55.0 |
| 85.0 | 65.0 | **86.7** | **88.3** | 51.7 | 66.7 |
| **91.7** | **93.3** | 85.0 | **88.3** | **61.7** | 71.7 |

## 4.1 BENCHMARKS, BASELINES AND METRICS

**Benchmarks.** LIBERO-LONG (Liu et al., 2024) encompasses diverse object interactions and versatile motor skills. We pre-train our model on the LIBERO-90 dataset, which includes demonstrations for 90 short-horizon tasks with **full annotations**, and then fine-tune and evaluate it on LIBERO-LONG, which features long-horizon tasks. CALVIN ABC-D (Mees et al., 2022) is a benchmark focusing on language-conditioned visual robot manipulation. It contains 34 tasks across four distinct environments (Env A, B, C, and D), each varying in object and scene visual appearance. For pre-training, we utilize the official robot play data with **no language instructions**, while the remaining data with full annotations is used for fine-tuning.

**Baselines.** For LIBERO-LONG, we implement a vanilla multi-task policy MTACT without pre-training, general image-based pre-trained policy MVP (Xiao et al., 2022), video-based pre-trained policy MPI (Zeng et al., 2024) and robot-data-based pre-trained policy OpenVLA (Kim et al., 2024). For CALVIN ABC-D, we select baselines that have demonstrated top competitive performance in prior reports. Roboflamingo (Li et al., 2023) is a method stepped from a vision-language model (Alayrac et al., 2022). Susie (Black et al., 2023) and CLOVER (Bu et al., 2024) are selected as the representative of two-stage PIDM methods. GR-1 (Wu et al., 2024) enhances manipulation with generative video pre-training, while the 3D Diffusor Actor (Ke et al., 2024) specializes in capturing 3D representations to enhance manipulation.

**Metrics.** In LIBERO-LONG, each method is evaluated across 20 rollouts with varying initial states for each task. We report both per-task and average success rates. In CALVIN ABC-D, the robot executes 1,000 task sequences, with each sequence comprising five consecutive tasks. We report the average success rates and the average length of completed sequences.

## 4.2 SIMULATION MAIN RESULTS

We conduct experiments on the LIBERO-LONG benchmark. As presented in Table 1, our policy achieved an average success rate of 78.7% without pre-training. After pre-training, the success rate increases by 9%, significantly outperforming the baselines. Compared to MTACT, our policy is more effective and benefits further from pre-training. The visual pre-training methods MVP and MPI achieve performance levels only comparable to our policy without pre-training, suggesting that visual pre-training alone is insufficient for manipulation tasks. Pre-training the entire policy using robotic data is necessary to enhance both perception and decision-making capabilities. In comparison to OpenVLA, Seer (316M total parameter, 65M trainable parameter) uses only 4% of its parameters (7B) yet achieves a 62% relative improvement in performance. These results underscore the advantages Seer and demonstrate the effectiveness of our pre-training objectives.

Table 2: **CALVIN ABC-D results.** We present the average success rates of top-3 checkpoints computed over 1000 rollouts for each task and the average number of completed tasks to solve 5 instructions consecutively (Avg. Len.). Seer shows consistent and significant superiority over baselines. The best results are **bolded**.

| Method | Task completed in a row | | | | | Avg. Len. ↑ |
|---|---|---|---|---|---|---|
| | 1 | 2 | 3 | 4 | 5 | |
| Roboflamingo | 82.4 | 61.9 | 46.6 | 33.1 | 23.5 | 2.47 |
| Susie | 87.0 | 69.0 | 49.0 | 38.0 | 26.0 | 2.69 |
| GR-1 | 85.4 | 71.2 | 59.6 | 49.7 | 40.1 | 3.06 |
| 3D Diffusor Actor | 92.2 | 78.7 | 63.9 | 51.2 | 41.2 | 3.27 |
| CLOVER | 96.0 | 83.5 | 70.8 | 57.5 | 45.4 | 3.53 |
| Seer (scratch) | 93.0 | 82.4 | 72.3 | 62.6 | 53.3 | 3.64 |
| Seer | 94.4 | 87.2 | 79.9 | 72.2 | 64.3 | 3.98 |
| Seer-Large (scratch) | 92.7 | 84.6 | 76.1 | 68.9 | 60.3 | 3.83 |
| **Seer-Large** | **96.3** | **91.6** | **86.1** | **80.3** | **74.0** | **4.28** |

We evaluate various methods on CALVIN ABC-D. Evaluation is conducted in Environment D, which differs visually from Environments A, B, and C where the training data was collected. Notably, the pre-training data lacks language annotations and includes meaningless actions and random explorations within these environments. As shown in Table 2, our method significantly outperforms the baselines. Our method surpasses the two-stage PIDM Susie (Wu et al., 2024) by a large margin, and is markedly superior to CLOVER (Bu et al., 2024), probably due to our delicate model design and end-to-end training paradigm. It also outperforms the video generative pre-training method GR-1 (Wu et al., 2024), demonstrating the advantage of end-to-end joint pre-training of visual prediction and action execution. Moreover, the results reveal that our pre-training method delivers considerable gains even on datasets lacking language annotations and containing noise, highlighting its adaptability and flexibility. The Seer-Large variant achieves Avg. Len. of **4.28**, establishing a new state-of-the-art on CALVIN ABC-D benchmark.

## 4.3 DATA EFFICIENCY

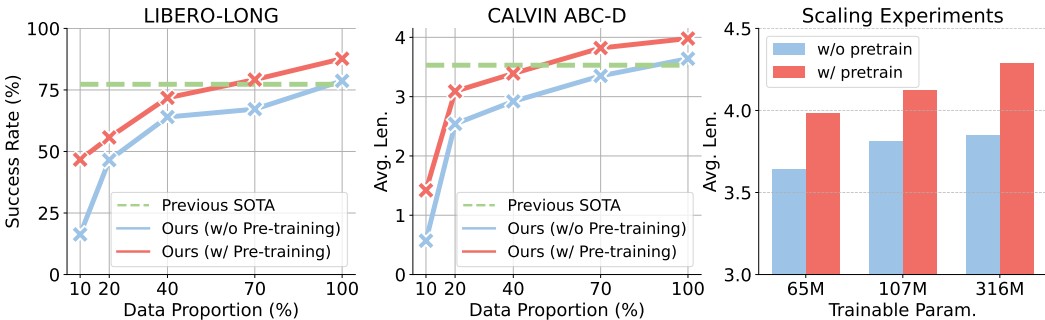

Figure 3: **Data efficiency and Scalability.** The two figures on the left depict Seer's performance on LIBERO and CALVIN using different proportions of the downstream data. And the figure on the right illustrates Seer's performance on CALVIN with various model sizes.

Collecting robot data is both time-consuming and labor-intensive, making data efficiency crucial for robot learning. We evaluate our method on two benchmarks: LIBERO-LONG and CALVIN ABC-D, using 10%, 20%, 40%, 70%, and 100% of the available data to fine-tune pre-trained policies or to train policies from scratch. The results, shown in Figure 3, demonstrate that our method consistently enhances policy performance across varying data sizes. Notably, under data-scarce conditions with only 10% of the training data, the pre-trained policy achieves a 187% relative improvement in success rate on LIBERO-LONG and a 150% relative improvement in average task length on CALVIN ABC-D compared to training from scratch. Additionally, our method only requires 70% data on LIBERO-LONG and on CALVIN ABC-D respectively to surpass state-of-the-art baselines. These results highlight the potential of Seer in scenarios with limited fine-tuning data.

Table 3: **Ablation studies on fine-tuning and pre-training objectives.** Integrating the conditional visual foresight objective $\mathcal{L}_{\text{fore}}$ and inverse dynamics prediction objective $\mathcal{L}_{\text{inv}}$ yields the best performance among pre-training and fine-tuning.

(a) **Fine-tuning objectives.**

| Fine-tuning $\mathcal{L}_{\text{fore}}$ | $\mathcal{L}_{\text{inv}}$ | 1 | 2 | 3 | 4 | 5 | Avg. Len. |
|---|---|---|---|---|---|---|---|
| × | × | 89.9 | 77.6 | 64.6 | 54.4 | 44.8 | 3.31 |
| ✓ | × | 91.2 | 78.6 | 67.1 | 56.6 | 47.8 | 3.41 |
| ✓ | ✓ | 93.0 | 82.4 | 72.3 | 62.6 | 53.3 | 3.64 |

(b) **Pre-training objectives.**

| Pre-train $\mathcal{L}_{\text{fore}}$ | $\mathcal{L}_{\text{inv}}$ | 1 | 2 | 3 | 4 | 5 | Avg. Len. |
|---|---|---|---|---|---|---|---|
| × | × | 93.0 | 82.4 | 72.3 | 62.6 | 53.3 | 3.64 |
| ✓ | × | 92.3 | 83.0 | 74.2 | 65.9 | 57.5 | 3.73 |
| ✓ | ✓ | 94.4 | 87.2 | 79.9 | 72.2 | 64.3 | 3.98 |

## 4.4 SCALABILITY

We evaluate Seer with different model size on the CALVIN ABC-D benchmark. Their results for the "Avg. Len." metrics are presented in Figure 3. It is observed that, regradless of the involvement of pre-training, the performances improve with the increase in trainable parameters, showcasing the scalability of Seer.

## 4.5 ABLATION STUDIES

We investigate the contributions of conditional visual foresight objective $\mathcal{L}_{\text{fore}}$ and inverse dynamics prediction objective $\mathcal{L}_{\text{inv}}$ during pre-training and fine-tuning on CALVIN ABC-D. The objectives during the fine-tuning phase are most closely related to performance in downstream tasks. Thus, we prioritize ablating the fine-tuning objectives before examining the pre-training objectives.

**Fine-tuning objectives.** We study the importance of $\mathcal{L}_{\text{fore}}$ and $\mathcal{L}_{\text{inv}}$ during fine-tuning. As shown in Table 3a, compared to the vanilla baseline, which directly performs behavioral cloning (w/o $\mathcal{L}_{\text{fore}}$, w/o $\mathcal{L}_{\text{inv}}$), separately predicting additional future images (w/ $\mathcal{L}_{\text{fore}}$, w/o $\mathcal{L}_{\text{inv}}$) yields improvements. This indicates the benefits of involving future image predictions (Wu et al., 2024). More importantly, integrating $\mathcal{L}_{\text{fore}}$ and $\mathcal{L}_{\text{inv}}$ results in a further boost in performance. This demonstrates that utilizing visual expectations to guide action predictions is a more effective strategy for leveraging the rich visual and temporal information inherent in robot data than the ablated version (w/ $\mathcal{L}_{\text{fore}}$, w/o $\mathcal{L}_{\text{inv}}$).

**Pre-training objectives.** Once the fine-tuning objectives ($\mathcal{L}_{\text{fore}} + \mathcal{L}_{\text{inv}}$) are established, we start to ablate pre-training objectives. The results in Table 3b indicate that pre-training only the vision prediction module (w/ $\mathcal{L}_{\text{fore}}$, w/o $\mathcal{L}_{\text{inv}}$) yields certain benefits, probably due to the vision priors learned from the extensive data. Moreover, pre-training the whole policy (w/ $\mathcal{L}_{\text{fore}}$, w/ $\mathcal{L}_{\text{inv}}$) through the integration of visual foresight and inverse dynamics results in greater improvements. This underscores the importance of the synergy between action and vision priors distilled from large robot datasets in enhancing performance on downstream tasks.

# 5 REAL-WORLD EXPERIMENTS

We evaluated Seer on six real-world tasks, including four focused on generalization and two focused on high precision and rich contact. We aim to answer: 1) Is Seer effective in real-world tasks? 2) Does pre-training consistently improve performance under intense disturbances?

## 5.1 REAL-WORLD BENCHMARK

**Real-world Setup.** We evaluate on a Franka Research 3 robot with a Robotiq-2f-85 gripper across six tasks, using two RealSense D435i cameras configured as Eye-on-Hand and Eye-on-Base for visual input. Four generalization-centric tasks are shown in Figure 4, and two high-precision and contact-rich tasks are presented in Figure A-3 in the appendix.

**Datasets.** For pre-training, we select the DROID dataset with Franka robots performing tasks in varied scenes. In the fine-tuning phase, we capture RGB images, robot states, and actions at 15 Hz, collecting 100 demonstrations per task.

**Baselines and Metrics.** We benchmark against MVP (image-based pre-trained), MPI (video-based pre-trained), and OpenVLA (robot-data-based pre-trained). Each method is evaluated over 15 trials

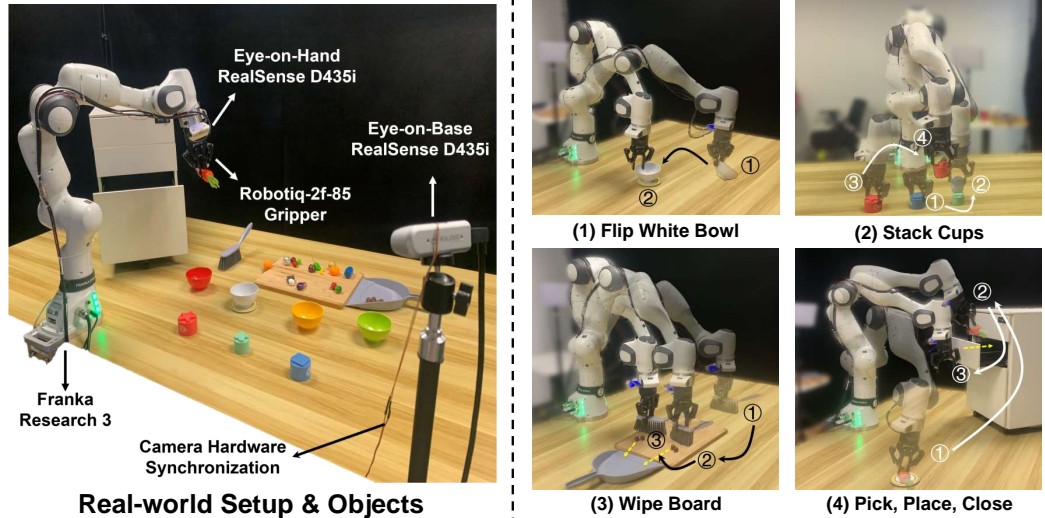

Figure 4: **Real-world Benchmark of four generalization-centric tasks. Left:** We use a Franka Research 3 robot with a Robotiq-2f-85 gripper and two RealSense D435i cameras. **Right:** We design four real-world manipulation tasks: (1) Flip White Bowl, (2) Stack Cups, (3) Wipe Board, (4) Pick, Place, Close. The detailed description of these tasks are included in A.6.1.

per task, with variations in the initial states of the objects. Each method is allowed three executions per trial, with the mean performance reported. Given the long-horizon and challenging nature of the tasks, we define two metrics: **Success Rate (SR)** and **Score** (as referenced in (Kim et al., 2024)). The **Score** accumulates during the completion of specific intermediary stages, while **SR** is recorded as 100% only upon successful completion of the entire task. Details are available in the Appendix.

## 5.2 REAL-WORLD MAIN RESULTS

Table 4: **Real-world main results.** We evaluate all the methods with 15 (cases) × 3 (repeated trials) rollouts per task. Our method achieves better performances among all tasks than baselines.

| Method | Demos Per Task | SR (%) ↑ / Score ↑ | | | | |
| --- | --- | --- | --- | --- | --- | --- |
| | | Flip White Bowl | Stack Cups | Wipe Board | Pick, Place, Close | Avg. |
| MVP | 100 | 80.0 / 24.0 | 26.7 / 26.0 | 53.3 / 38.0 | 60.0 / 31.0 | 55.0 / 29.8 |
| MPI | 100 | 66.7 / 21.0 | 26.7 / 29.0 | 33.3 / 35.0 | 66.7 / 32.0 | 48.4 / 29.3 |
| OpenVLA | 100 | 53.3 / 19.0 | 0.00 / 8.00 | 0.00 / 4.00 | 13.3 / 13.0 | 16.7 / 11.0 |
| Seer (scratch) | 100 | 60.0 / 19.0 | 46.7 / 35.0 | 60.0 / 37.0 | 73.3 / 40.0 | 60.0 / 32.8 |
| Seer | 100 | **86.7 / 26.0** | **60.0 / 42.0** | **73.3 / 41.0** | **86.7 / 42.0** | **78.4 / 39.5** |

As can be seen in Table 4, our pre-trained policy could outperform all the baselines over all tasks. Specifically, our method improves the average success rate and the accumulated score from 60.0% to 78.4% and from 32.8 to 39.5 compared to the version trained from scratch. In comparison with MVP and MPI, which only pre-train vision encoders, our results reinforce the importance of pre-training the entire policy, aligning with findings from simulation experiments. Regarding the performance of OpenVLA in the real world, it has a significantly larger tunable model size (3B here) during full fine-tuning and relies solely on an eye-on-base camera. This could lead to severe overfitting and coarse action predictions, particularly in tasks where objects are small (as in Stack Cups) or located far from the camera (as in Wipe Board). In contrast, our method demonstrates better handling of these tasks due to its moderate model size and comprehensive data utilization. We also conduct evaluation on two high-precision and contact-rich tasks, with the results presented in Table A-III.

## 5.3 ROBUSTNESS

We propose several generalization types to assess the effectiveness of our pre-trained policy across multiple settings. As shown in Figure 5, in **Flip White Bowl**, we introduce bowls of different colors alongside the original white bowl. These bowls share identical shape, size, and material, which

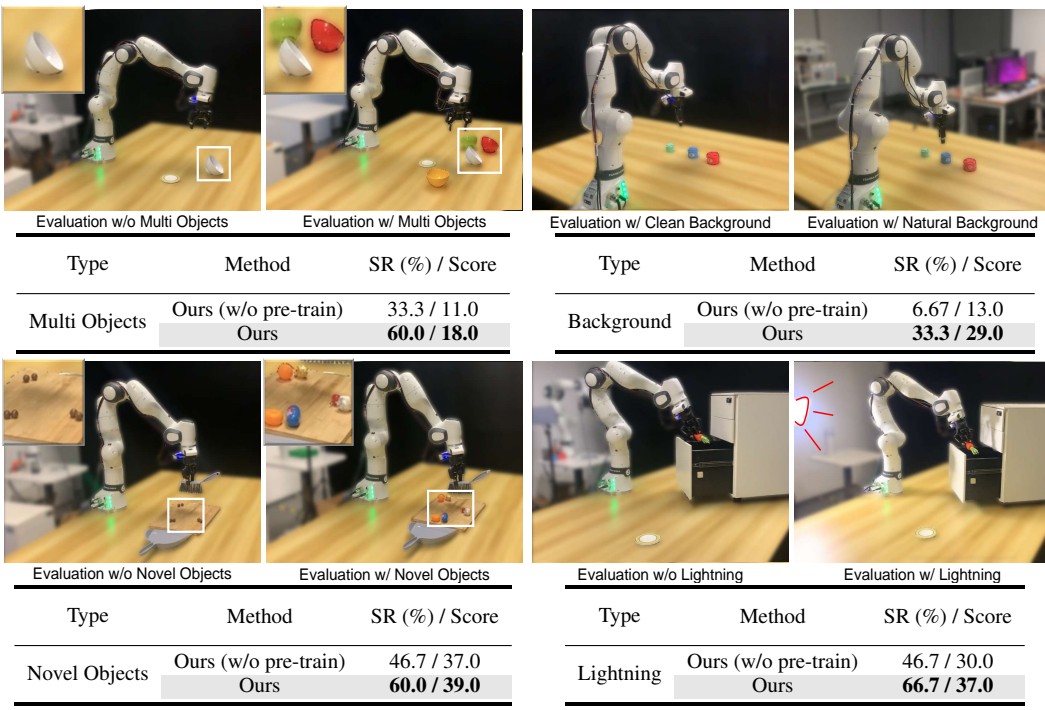

Figure 5: **Generalization evaluation.** We design a generalization test per task with different disturbances. **Top Left:** In Flip Bowl, we put several bowls with the same shape, size, material and different colors around the original white one. **Top Right:** In Stack Cups, we remove the original black backdrop and keep the natural background. **Bottom Left:** In Wipe Board, we replace the chocolate balls with diverse novel small objects. **Bottom Right:** In Pick, Place, Close, we introduce an additional light source. Among all tests, our pre-trained method brings consistent benefits.

could potentially mislead the algorithm. In **Wipe Board**, we replace the original chocolate balls with novel objects that vary in mass, shape, and coefficients of friction, thereby increasing the task's difficulty. In both scenarios, our pre-trained policy demonstrates significant improvements in success rate (SR) and Score. We attribute these enhancements to the extensive variety of interactable and distractible objects present in the pre-training dataset, which strengthens the model's semantic understanding. Additionally, in **Pick, Place, Close**, we incorporate a strong light source that alters the visual appearance of objects in RGB images. In **Stack Cups**, we remove the clean black backdrop and replace it with a natural background, introducing complex disturbances such as variable lighting, unseen distractions, and effects on camera exposure. Even under these challenging conditions, our pre-trained policy continues to deliver satisfactory results. These results demonstrate that extensive pre-training on large-scale robot datasets with diverse scenes enhances robustness.

## 6   CONCLUSION AND LIMITATIONS

In this work, we introduce Seer, an end-to-end predictive inverse dynamics model that synergizes conditional visual foresight with inverse dynamics prediction for manipulation. Seer achieves state-of-the-art results on two simulation benchmarks, and shows significant improvements and robustness in real-world experiments after pre-training on the large DROID robot dataset. The limitations mainly lie in two aspects. Firstly, we only evaluate six downstream tasks. A broader spectrum of high-precision and contact-rich tasks remains to be explored. Secondly, evaluating across different robots is also necessary to test Seer's cross-embodiment capability.

## ACKNOWLEDGMENTS

This work is supported by the National Key R&D Program of China (2022ZD0160201), Shanghai Artificial Intelligence Laboratory, and China Postdoctoral Science Foundation (2023M741848).

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

# A APPENDIX

## A.1 IMPLEMENTATION DETAILS

In this section, we outline the implementation details of our framework.

**Vision.** We employ a MAE-pretrained ViT-B (He et al., 2022) as the vision encoder. At each timestep, images are captured from two views: eye-on-hand and eye-on-base. Each image is processed by the vision encoder to produce 196 latent vectors, which represent local patch information, along with a [CLS] token that encodes the global representation of the image. Directly inputting all 197 tokens into the transformer backbone would create a significant computational burden, particularly when processing long histories. Moreover, many image details are redundant for accomplishing manipulation tasks. To address this, we utilize the Perceiver Resampler (Alayrac et al., 2022) to condense the image representations and extract task-relevant features. The Perceiver Resampler employs learnable latent vectors with a shape of (num_latents, dim), where num_latents is significantly smaller than the number of image tokens. Through Perceiver Attention, these latent vectors condense the input image features, along with the [CLS] token, to form the final image tokens.

**Robot state.** The robot state consists of the arm state and the gripper state. The arm state includes the end-effector position and its rotation in Euler angles, resulting in a six-dimensional representation. The gripper state is a binary value indicating whether the gripper is open or closed. We tokenize the robot state using an MLP. Specifically, the gripper state is first converted into a one-hot encoding. The one-hot encoding of the gripper state and the arm state are then each passed through separate linear layers. The outputs are concatenated and passed through a final linear layer to produce the state token.

**Language.** Text instructions are encoded using the CLIP ViT-B/32 text encoder (Radford et al., 2021) and projected through a linear layer to generate the language token.

**Readout tokens.** At each timestep, we append $[FRS]$ and $[INV]$ tokens to read out foresight and actions. Specifically, $[FRS]$ tokens are appended to extract representations for two views, and three $[INV]$ tokens are appended to predict actions across three steps, ensuring temporal action consistency and robustness to idle actions.

**Decoder.** After passing through the transformer backbone, the action and image latents generated by the [INV] and [FRS] tokens are input to the action decoder and image decoder to predict actions and images for conditional visual foresight and inverse dynamics prediction.

The action decoder is an MLP that decodes the action latent into a seven-dimensional action vector. First, the action latent is processed by a linear layer followed by a ReLU activation function. Then, it passes through a second linear layer with ReLU activation. The output is fed into two independent decoders: the arm action decoder and the gripper action decoder. The arm action decoder maps the high-dimensional vector to a six-dimensional output through a linear layer, applying a Tanh activation function to constrain the arm action within the range [-1, 1]. The gripper action decoder also employs a linear layer to map the latent vector to a one-dimensional output, applying a Sigmoid activation function to constrain the gripper action between [0, 1]. A gripper action value of 0.5 or higher is interpreted as 1 (closed), while a value below 0.5 is interpreted as 0 (open).

For image decoding, following (He et al., 2022), we use a vision transformer (ViT) as the image decoder. The image decoder receives the image latent and mask tokens from the transformer backbone as input. Positional information is provided through fixed sine-cosine positional encodings. The inputs are processed by multiple transformer encoder blocks. Finally, a linear layer predicts the pixels for each patch, generating the image that represents the predicted future state.

We present relevant hyperparameters during both pretraining and finetuning in Table A-I.

The standard version of Seer contains 316M parameters, where only 65M is tunable. For all simulation results, we use eight 4090 GPUS to pre-train and fine-tune. The pre-training process requires about 40 hours for CALVIN ABC-D and and 30 hours for LIBERO-LONG. The fine-tuning process requires about 24 hours for CALVIN ABC-D and 6 hours for LIBERO-LONG.

Table A-I: Training hyperparameters.

| Hyperparameters | Pre-training | Fine-tuning |
|---|---|---|
| Batch Size | 640 (LIBERO & CALVIN) / 2048 (Real) | 512 |
| Learning Rate | 1e-4 | 1e-3 |
| Optimizer | AdamW | AdamW |
| Learning Rate Schedule | Cosine decay | Cosine decay |
| Training Epochs | 30 (LIBERO & Real) / 20 (CALVIN) | 40 (LIBERO & Real) / 20 (CALVIN) |
| History Length | 7 (LIBERO & Real) / 10 (CALVIN) | 7 (LIBERO & Real) / 10 (CALVIN) |
| Action Chunk Length | 3 | 3 |

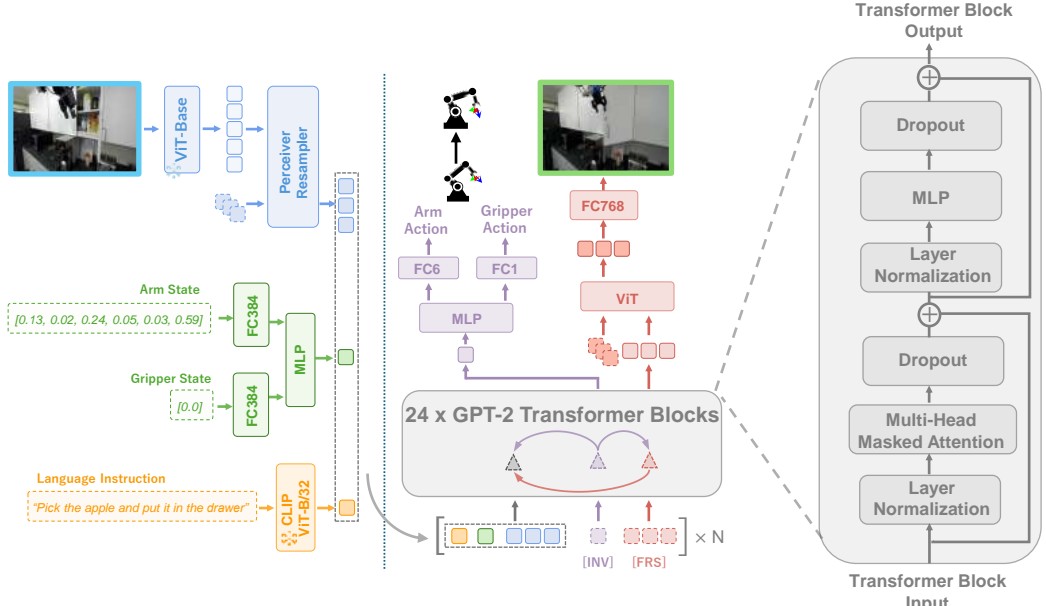

Figure A-1: Network Architecture.

Table A-II: Hyperparameters for the transformers in our policy.

| | Hidden size | Number of layers | Number of heads |
|---|---|---|---|
| image encoder | 768 | 12 | 12 |
| perceiver resampler | 768 | 3 | 8 |
| transformer backbone | 384 | 24 | 12 |
| image decoder | 384 | 2 | 16 |

## A.2 Network Architecture

As presented in Figure A-1, Seer consists of the following modules: image encoder, perceiver resampler, robot state encoder, language encoder, transformer backbone, action decoder and image decoder. Here we describe each module in detail:

- image encoder: a MAE pre-trained ViT-Base (He et al., 2022) encoder. Details can be seen in Table A-II.
- perceiver resampler: a module to compress the image tokens efficiently. Details can be seen in Table A-II.
- robot state encoder: MLPs that encode robot states into a latent space.
- language encoder: a CLIP (Radford et al., 2021) ViT-B/32 text encoder.
- transformer backbone: 24 layers of GPT-2 transformer blocks, with a hidden size of 384 and 12 heads. It takes image tokens, language tokens, robot states tokens, [INV], [FRS] as inputs. Details can be seen in Table A-II.
- action decoder: MLPs that decode the latent feature into 7-DOF action.
- image decoder: a ViT-based transformer followed by a linear layer. Details can be seen in Table A-II.

## A.3 Baseline Implementation

In the simulation benchmark, we report the scores for Roboflamingo, Susie, GR-1, and the 3D Diffusor Actor from their respective papers. For MTACT and OpenVLA, we reproduce the results using the official code. For MVP and MPI, we replace the vision encoder in our policy with their pretrained versions. Thanks to the strong design of our policy, MVP and MPI show competitive performance, though they only approach the results of our policy without pretraining.

## A.4 LIBERO-Long Experiment details

LIBERO (Liu et al., 2024) is a novel benchmark for lifelong learning in robot manipulation, comprising four task suites: LIBERO-SPATIAL, LIBERO-OBJECT, LIBERO-GOAL, and LIBERO-100. The first three task suites are designed to disentangle the transfer of declarative and procedural knowledge, while LIBERO-100 consists of 100 tasks involving entangled knowledge transfer. LIBERO-100 includes 100 tasks that require diverse object interactions and versatile motor skills. LIBERO-100 is divided into 90 short-horizon tasks (LIBERO-90) and 10 long-horizon tasks (LIBERO-LONG). We use LIBERO-90 as the pretraining dataset, while LIBERO-LONG is utilized for the downstream finetuning and evaluation.

The policy use images from both fixed and gripper cameras to observe the environment, which were resized to 224x224 pixels. We also incorporated the robot state to help the policy understand the robot's self-state, including the position and orientation of the end effector and the gripper state indicating the width between the grippers. The action space consists of a seven-dimensional vector: six dimensions represent arm actions, and one dimension indicates the gripper's open/close state. The arm action represents the 6D pose (position and orientation) of a controlled frame. This frame is located between the fingers of robots.

## A.5 CALVIN ABC-D Experiment details

CALVIN (Mees et al., 2022) is a simulated benchmark designed for learning long-horizon, language-conditioned tasks. Its goal is to enable the development of agents capable of solving various robotic manipulation tasks using only onboard sensors and instructions provided in natural human language. The tasks in CALVIN are complex, involving long sequences and intricate language instructions. It supports flexible configurations of sensor suites. Agents are evaluated in a zero-shot manner on novel language instructions and unfamiliar environments.

The CALVIN benchmark includes four distinct but structurally similar environments–Env A, B, C and D. Each environment features a Franka Emika Panda robot arm equipped with a parallel gripper, as well as a desk with a sliding door and a drawer that can be opened and closed. And

there are several objects, such as blocks and buttons, on the desk. To more effectively assess the generalization of the learned policies, each environment features distinct textures, and objects like the sliding door, drawer, and light button, are placed in different positions.

CALVIN offers rich observations for robot learning. We use images from both fixed and gripper-mounted cameras, resized to 224x224 pixels, along with robot state information, which includes end-effector position, orientation, and gripper state (open/close). The action space is a 7-dimensional vector: six dimensions correspond to end-effector displacement (position and orientation), and one dimension controls the gripper's open/close state. The unstructured data includes exploratory and sub-optimal behaviors, comprising approximately 2.4 million interaction steps and 40 million short-horizon windows. Data from Env A, B, and C, which lacks language annotations, is used to pretrain the policy, while data with language annotations is used for downstream task learning. Env D is reserved for policy evaluation.

## A.6 REAL WORLD EXPERIMENT DETAILS

### A.6.1 GENERALIZATION-CENTRIC TASKS

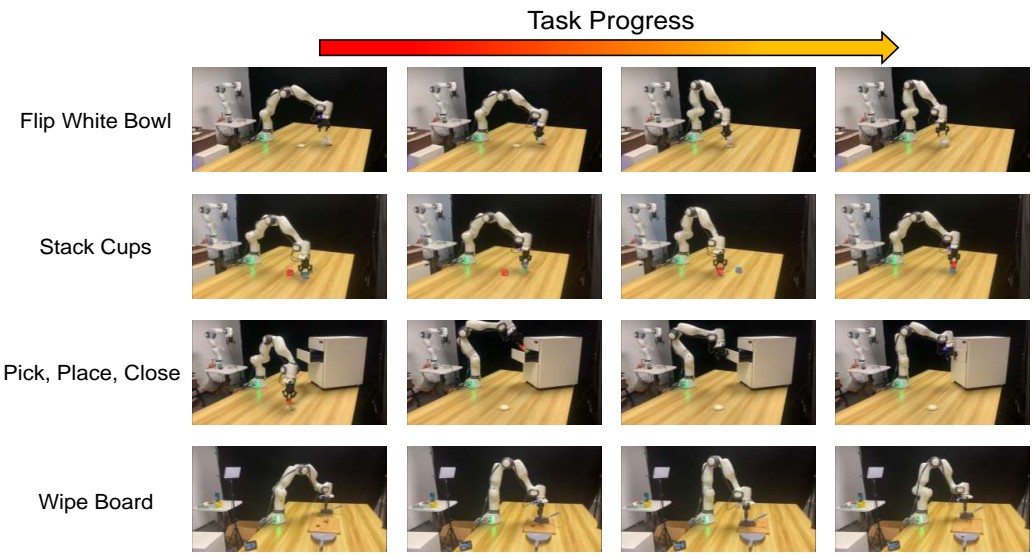

Figure A-2: Progress and configuration of generalization-centric tasks.

**Tasks.** In **Flip White Bowl**, the robot picks up a randomly overturned bowl and places it on a coaster, testing basic 6DoF pick & place capability. In **Stack Cups**, the robot stacks three randomly placed cups of different colors and sizes. It firstly covers the small cup with the middle one, and then covers the middle cup with the big one. Since the cup surface is smooth and covering requires a close fit, this task challenges fine-grained action predictions. In **Wipe Board**, the robot collects 3 to 7 chocolate balls arranged in 1 to 3 clusters. It uses a brush to sweep the balls into a dustpan, testing its ability to handle multi-modal settings and perform repetitive motions. In **Pick, Place, Close**, the robot picks a randomly set carrot, transports it into an opened drawer and closes the drawer. This evaluates skills of executing consecutive actions in a large space with articulated objects.

**Flip White Bowl:** the robot needs to ①pick an overturned bowl and ②place it on the coaster. In this task, the white bowl is randomly placed on the table within a 40cm × 40cm square, and the coaster is randomly placed within a 15cm × 15cm square. The robot needs to pick up the white bowl and put it on the coaster. In the generalization test, 1 to 3 bowls with identical shapes, sizes and materials are added to disturb the policy. Success rate (SR) is recorded as 100% only when the white bowl is placed on the coaster safely. If the bowl is successfully grasped, the score will plus one (+1). If the bowl is successfully placed on the coast, the score will also plus one (+1). The full score in this task is 2.

**Stack Cups:** the robot needs to ①pick the middle cup, ②cover the small one, ③pick the big one, and ④cover the middle one. In this task, three cups of different sizes are randomly placed on the table within a 40cm × 40cm square. The robot needs to cover the small cup with the middle one, and cover the middle cup with the big one. Only when all the cups are stacked precisely and in a correct order will the SR be recorded as 100%. The score will plus one (+1) when each primitive action (pick or place) is accomplished. The full score in this task is 4.

**Pick, Place, Close:** the robot needs to ①pick the carrot, ②put it in the drawer, and ③close the drawer. In this task, a drawer with three layers is fixed on the table. During each test, one of three layers is open. A carrot on the coaster is randomly placed within a 20cm × 20cm square. The orientation of the carrot is randomized. The robot needs to pick the carrot, place it into a certain layer, and close the drawer. The score will plus one (+1) when successfully (1) picking the carrot, (2) placing the carrot, and (3) closing the drawer. The full score in this task is 3.

**Wipe Board:** the robot needs to ①grasp the brush, and ②③sweep all the chocolate balls into the dustpan. In this task, a board (30cm × 40cm) and dustpan is fixed on the table. A brush is randomly placed within a 5cm × 5cm square. 3 to 7 chocolate balls (diameter 1cm) are divided into 1 to 3 clusters and randomly placed on the whole board. Only when all the chocolate balls are swept into the dustpan safely will the SR be recorded as 100%. The score will plus one (+1) when (1) grasping the brush successfully, (2) sweeping partial chocolates into the dustpan, and (3) sweeping all chocolates into the dustpan. The full score in this task is 3.

### A.6.2 HIGH-PRECISION AND CONTACT-RICH TASKS

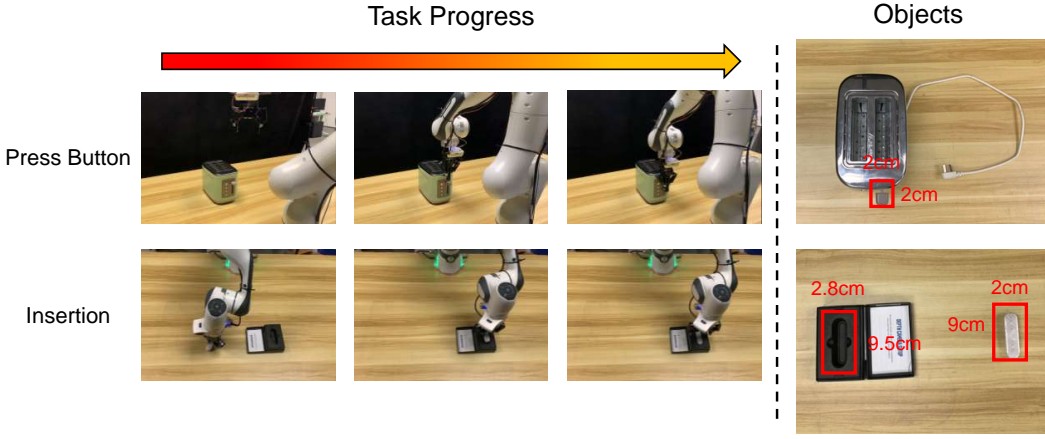

Figure A-3: Process and configuration of high-precision and contact-rich tasks.

Table A-III: Results on high-precision and contact-rich tasks.

| Method | Demos Per Task | Press Button SR (%) ↑ / Score ↑ | Insertion SR (%) ↑ / Score ↑ |
|---|---|---|---|
| MVP | 100 | 46.7 / 17.0 | 26.7 / 11.0 |
| Ours (w/o pre-train) | 100 | 40.0 / 13.0 | 40.0 / 16.0 |
| **Ours** | **100** | **60.0 / 18.0** | **60.0 / 19.0** |

**Press Button:** In this task, the toaster is randomly placed in a 30cm × 30cm square. The robot is required to approach the toaster, close the fingers, push the button from a top-down view, and exceed 3/4 of the scale (Figure A-3). The score will plus one (+1) when (1) pushing the button successfully with no collision, and (2) exceeding 3/4 of the scale. The full score in this task is 2.

**Insertion:** In this task, a 2cm × 9cm camera model is randomly placed in a 20cm × 20cm square. The robot is required to pick the camera model and insert it into a 2.8cm × 9.5cm groove without

any collision (Figure A-3). The score will plus one (+1) when (1) grasping the camera model, and (2) inserting successfully with no collision. The full score in this task is 2.

**Results:** As seen in Table A-III, pre-training on the DROID dataset brings obvious improvements, compared to the scratch version and previous state-of-the-art baselines. Notably, both tasks require quite precise action predictions and collision-free interactions, showing our model's potential in high-precision and contact-rich tasks.

### A.6.3 REAL-WORLD IMPLEMENTATION DETAILS

During real-world training, we set the sequence length as 7, visual foresight steps and action prediction steps as 3. We utilize the MAE pre-trained ViT-B encoder and configured it as bfloat16 to accelerate inference. We find this quantization won't produce side effects on manipulation tasks. The pre-training dataset, e.g., DROID involves 76K successful trajectories, and the downstream fine-tuning data involves 400 demos. Key hyperparameters are listed in Table A-I. We utilize the 9-th pre-trained checkpoint for fine-tuning, and evaluate performance using the 17-th fine-tuned checkpoint.

In real-world experiments involving the MVP and MPI baselines, we replace the MAE pre-trained vision encoder in our network with the MVP pre-trained and MPI pre-trained counterparts, respectively. We then fine-tune these two baselines on the downstream tasks and report performances. For OpenVLA, we choose to fully finetune its public released checkpoint with model size 7B pre-trained on OXE. The fine-tuning config is identical to the one trained on Bridge dataset in OpenVLA's public codebase. We fine-tune this large model using eight A100 GPUs with more than 24 hours and use the checkpoint with lowest average validation loss to evaluate.

### A.6.4 ACROSS EMBODIMENTS EXPERIMENTS

To investigate the impact of Seer on cross-embodiment, we conduct an additional real-world experiment, where Seer is pre-trained on OXE and fine-tuned on self-collected data. We refer the subset mix-up recipe in Octo (Ghosh et al., 2024), remove all the subset that includes franka robots, filter subsets with odd action labels, and save the rest subsets as OXE pre-training dataset.

Table A-IV: **Results of across embodiments experiments.**

| Method | SR (%) ↑ / Score ↑ | | | | | | |
|---|---|---|---|---|---|---|---|
| | Flip White Bowl | Stack Cups | Wipe Board | Pick, Place, Close | Press Button | Insertion | Avg. |
| Seer (scratch) | 60.0 / 19.0 | 46.7 / 35.0 | 60.0 / 37.0 | 73.3 / 40.0 | 40.0 / 13.0 | 40.0 / 16.0 | 53.3 / 26.7 |
| Seer (OXE) | 73.3 / 22.0 | 40.0 / 30.0 | 66.7 / 40.0 | 80.0 / 41.0 | 33.3 / 10.0 | 46.7 / 18.0 | 56.7 / 26.8 |
| Seer (Droid) | **86.7 / 26.0** | **60.0 / 42.0** | **73.3 / 41.0** | **86.7 / 42.0** | **60.0 / 18.0** | **60.0 / 19.0** | **71.1 / 31.3** |

As seen in the Table A-IV, pre-training on the OXE dataset only provides marginal improvements in most tasks. Moreover, in some high-precision tasks, the OXE pre-trained version even brings negative effects. We attribute marginal improvements in general manipulation tasks to the diversity of objects, tasks, scenes, and language instructions in OXE. For the slight decrease in some high-precision tasks, we suspect two reasons: Firstly, for high-precision tasks, images from the eye-on-hand camera (wrist view) are quite important, since this camera captures a lot of local and detailed information for policy. However, in most OXE subsets, only images from third-person view with diverse camera poses are provided, with most wrist view images missing. Secondly, the gap in cross-embodiments and cross-action-controller types exists in our settings. High-precision tasks require a concentrated and precise action distribution. However, pre-training on OXE may offer a distant action prior due to the aforementioned physics gap.

### A.6.5 DETAILED REAL-WORLD RESULTS

The raw records of the real-world experiments are shown in Table A-V, Table A-VI, Table A-VII, and Table A-VIII, which we use to calculate the success rate and score.

Table A-V: Detailed results (SR (%) / Score) in Flip White Bowl.

| Case Index | MVP | MPI | OpenVLA | Ours (w/o pre-train, 20 demos) | Ours (20 demos) | Ours (w/o pre-train, 100 demos) | Ours (100 demos) |
|---|---|---|---|---|---|---|---|
| 1 | 0.00 / 0.00 | 100 / 2.00 | 0.00 / 1.00 | 0.00 / 1.00 | 100 / 2.00 | 100 / 2.00 | 100 / 2.00 |
| 2 | 100 / 2.00 | 100 / 2.00 | 100 / 2.00 | 0.00 / 0.0 | 100 / 2.00 | 100 / 2.00 | 100 / 2.00 |
| 3 | 100 / 2.00 | 100 / 2.00 | 0.00 / 1.00 | 0.00 / 0.0 | 0.00 / 0.0 | 100 / 2.00 | 100 / 2.00 |
| 4 | 100 / 2.00 | 0.00 / 0.00 | 100 / 2.00 | 0.00 / 0.0 | 0.00 / 0.0 | 100 / 2.00 | 100 / 2.00 |
| 5 | 100 / 2.00 | 0.00 / 0.00 | 100 / 2.00 | 100 / 2.00 | 100 / 2.00 | 0.00 / 0.00 | 100 / 2.00 |
| 6 | 100 / 2.00 | 100 / 2.00 | 100 / 2.00 | 0.00 / 0.0 | 0.00 / 0.0 | 0.00 / 0.00 | 100 / 2.00 |
| 7 | 100 / 2.00 | 100 / 2.00 | 0.00 / 1.00 | 0.00 / 0.0 | 0.00 / 0.0 | 100 / 2.00 | 100 / 2.00 |
| 8 | 100 / 2.00 | 100 / 2.00 | 100 / 2.00 | 100 / 2.00 | 100 / 2.00 | 0.00 / 1.00 | 100 / 2.00 |
| 9 | 100 / 2.00 | 100 / 2.00 | 100 / 2.00 | 100 / 2.00 | 100 / 2.00 | 100 / 2.00 | 100 / 2.00 |
| 10 | 100 / 2.00 | 100 / 2.00 | 0.00 / 0.00 | 0.00 / 0.0 | 0.00 / 0.0 | 100 / 2.00 | 0.00 / 0.00 |
| 11 | 100 / 2.00 | 0.00 / 0.00 | 0.00 / 0.00 | 0.00 / 0.0 | 100 / 2.00 | 0.00 / 0.00 | 100 / 2.00 |
| 12 | 0.00 / 0.00 | 0.00 / 1.00 | 0.00 / 0.00 | 100 / 2.00 | 100 / 2.00 | 0.00 / 0.00 | 0.00 / 0.00 |
| 13 | 100 / 2.00 | 100 / 2.00 | 100 / 2.00 | 0.00 / 0.0 | 0.00 / 1.00 | 0.00 / 0.00 | 100 / 2.00 |
| 14 | 100 / 2.00 | 100 / 2.00 | 0.00 / 0.00 | 0.00 / 1.00 | 100 / 2.00 | 100 / 2.00 | 100 / 2.00 |
| 15 | 0.00 / 0.00 | 0.00 / 0.00 | 100 / 2.00 | 0.00 / 0.0 | 0.00 / 0.0 | 100 / 2.00 | 100 / 2.00 |

Table A-VI: Detailed results (SR (%) / Score) in Stack Cups.

| Case Index | MVP | MPI | OpenVLA | Ours (w/o pre-train, 20 demos) | Ours (20 demos) | Ours (w/o pre-train, 100 demos) | Ours (100 demos) |
|---|---|---|---|---|---|---|---|
| 1 | 0.00 / 3.00 | 100 / 4.00 | 0.00 / 2.00 | 0.00 / 0.00 | 0.00 / 1.00 | 100 / 4.00 | 100 / 4.00 |
| 2 | 100 / 4.00 | 0.00 / 3.00 | 0.00 / 0.00 | 0.00 / 0.00 | 0.00 / 0.00 | 0.00 / 1.00 | 100 / 4.00 |
| 3 | 0.00 / 3.00 | 0.00 / 3.00 | 0.00 / 0.00 | 0.00 / 1.00 | 0.00 / 0.00 | 0.00 / 2.00 | 100 / 4.00 |
| 4 | 100 / 4.00 | 0.00 / 1.00 | 0.00 / 1.00 | 0.00 / 1.00 | 0.00 / 0.00 | 100 / 4.00 | 100 / 4.00 |
| 5 | 0.00 / 1.00 | 0.00 / 3.00 | 0.00 / 2.00 | 0.00 / 1.00 | 0.00 / 1.00 | 100 / 4.00 | 0.00 / 2.00 |
| 6 | 0.00 / 1.00 | 100 / 4.00 | 0.00 / 0.00 | 0.00 / 1.00 | 0.00 / 2.00 | 100 / 4.00 | 100 / 4.00 |
| 7 | 0.00 / 0.00 | 100 / 4.00 | 0.00 / 0.00 | 0.00 / 1.00 | 0.00 / 0.00 | 0.00 / 1.00 | 100 / 4.00 |
| 8 | 0.00 / 0.00 | 0.00 / 0.00 | 0.00 / 1.00 | 0.00 / 0.00 | 0.00 / 0.00 | 0.00 / 0.00 | 0.00 / 0.00 |
| 9 | 100 / 4.00 | 100 / 4.00 | 0.00 / 0.00 | 100 / 4.00 | 0.00 / 0.00 | 100 / 4.00 | 100 / 4.00 |
| 10 | 0.00 / 1.00 | 0.00 / 1.00 | 0.00 / 0.00 | 0.00 / 0.00 | 0.00 / 0.00 | 0.00 / 0.00 | 0.00 / 2.00 |
| 11 | 0.00 / 0.00 | 0.00 / 1.00 | 0.00 / 0.00 | 0.00 / 1.00 | 0.00 / 1.00 | 0.00 / 0.00 | 0.00 / 0.00 |
| 12 | 0.00 / 0.00 | 0.00 / 0.00 | 0.00 / 0.00 | 0.00 / 0.00 | 0.00 / 0.00 | 0.00 / 0.00 | 0.00 / 0.00 |
| 13 | 0.00 / 0.00 | 0.00 / 0.00 | 0.00 / 1.00 | 0.00 / 0.00 | 0.00 / 0.00 | 100 / 4.00 | 0.00 / 2.00 |
| 14 | 0.00 / 1.00 | 0.00 / 0.00 | 0.00 / 1.00 | 0.00 / 0.00 | 0.00 / 1.00 | 0.00 / 0.00 | 100 / 4.00 |
| 15 | 100 / 4.00 | 0.00 / 1.00 | 0.00 / 0.00 | 0.00 / 1.00 | 0.00 / 1.00 | 100 / 4.00 | 100 / 4.00 |

Table A-VII: Detailed results (SR (%) / Score) in Pick, Place, Close.

| Case Index | MVP | MPI | OpenVLA | Ours (w/o pre-train, 20 demos) | Ours (20 demos) | Ours (w/o pre-train, 100 demos) | Ours (100 demos) |
|---|---|---|---|---|---|---|---|
| 1 | 100 / 3.00 | 100 / 3.00 | 0.00 / 0.00 | 100 / 3.00 | 100 / 3.00 | 100 / 3.00 | 100 / 3.00 |
| 2 | 100 / 3.00 | 100 / 3.00 | 100 / 3.00 | 0.00 / 0.00 | 100 / 3.00 | 100 / 3.00 | 100 / 3.00 |
| 3 | 100 / 3.00 | 100 / 3.00 | 0.00 / 0.00 | 0.00 / 2.00 | 100 / 3.00 | 0.00 / 2.00 | 100 / 3.00 |
| 4 | 0.00 / 0.00 | 100 / 3.00 | 0.00 / 0.00 | 0.00 / 0.00 | 0.00 / 0.00 | 0.00 / 2.00 | 100 / 3.00 |
| 5 | 0.00 / 0.00 | 100 / 3.00 | 0.00 / 2.00 | 100 / 3.00 | 100 / 3.00 | 100 / 3.00 | 100 / 3.00 |
| 6 | 0.00 / 1.00 | 0.00 / 0.00 | 0.00 / 1.00 | 0.00 / 1.00 | 0.00 / 1.00 | 0.00 / 2.00 | 100 / 3.00 |
| 7 | 100 / 3.00 | 100 / 3.00 | 0.00 / 0.00 | 0.00 / 0.00 | 100 / 3.00 | 100 / 3.00 | 100 / 3.00 |
| 8 | 100 / 3.00 | 100 / 3.00 | 0.00 / 0.00 | 100 / 3.00 | 100 / 3.00 | 100 / 3.00 | 100 / 3.00 |
| 9 | 0.00 / 1.00 | 0.00 / 0.00 | 100 / 3.00 | 0.00 / 0.00 | 0.00 / 0.00 | 0.00 / 1.00 | 0.00 / 2.00 |
| 10 | 100 / 3.00 | 100 / 3.00 | 0.00 / 0.00 | 0.00 / 0.00 | 0.00 / 2.00 | 100 / 3.00 | 100 / 3.00 |
| 11 | 100 / 3.00 | 0.00 / 0.00 | 0.00 / 1.00 | 0.00 / 0.00 | 0.00 / 1.00 | 100 / 3.00 | 100 / 3.00 |
| 12 | 0.00 / 1.00 | 0.00 / 1.00 | 0.00 / 1.00 | 0.00 / 1.00 | 0.00 / 0.00 | 100 / 3.00 | 0.00 / 1.00 |
| 13 | 100 / 3.00 | 100 / 3.00 | 0.00 / 0.00 | 100 / 3.00 | 100 / 3.00 | 100 / 3.00 | 100 / 3.00 |
| 14 | 100 / 3.00 | 100 / 3.00 | 0.00 / 2.00 | 0.00 / 0.00 | 0.00 / 0.00 | 100 / 3.00 | 100 / 3.00 |
| 15 | 0.00 / 1.00 | 0.00 / 1.00 | 0.00 / 0.00 | 0.00 / 0.00 | 100 / 3.00 | 100 / 3.00 | 100 / 3.00 |

Table A-VIII: Detailed results (SR (%) / Score) in Wipe Board.

| Case Index | MVP | MPI | OpenVLA | Ours (w/o pre-train, 20 demos) | Ours (20 demos) | Ours (w/o pre-train, 100 demos) | Ours (100 demos) |
|---|---|---|---|---|---|---|---|
| 1 | 100 / 3.00 | 100 / 3.00 | 0.00 / 0.00 | 0.00 / 0.00 | 0.00 / 2.00 | 100 / 3.00 | 100 / 3.00 |
| 2 | 100 / 3.00 | 100 / 3.00 | 0.00 / 0.00 | 0.00 / 2.00 | 100 / 3.00 | 0.00 / 2.00 | 100 / 3.00 |
| 3 | 100 / 3.00 | 100 / 3.00 | 0.00 / 1.00 | 100 / 3.00 | 100 / 3.00 | 0.00 / 0.00 | 100 / 3.00 |
| 4 | 100 / 3.00 | 0.00 / 2.00 | 0.00 / 0.00 | 100 / 3.00 | 0.00 / 2.00 | 100 / 3.00 | 100 / 3.00 |
| 5 | 0.00 / 2.00 | 0.00 / 2.00 | 0.00 / 1.00 | 0.00 / 2.00 | 100 / 3.00 | 100 / 3.00 | 100 / 3.00 |
| 6 | 0.00 / 2.00 | 0.00 / 2.00 | 0.00 / 0.00 | 0.00 / 2.00 | 0.00 / 2.00 | 0.00 / 2.00 | 0.00 / 2.00 |
| 7 | 0.00 / 2.00 | 0.00 / 2.00 | 0.00 / 0.00 | 0.00 / 1.00 | 0.00 / 1.00 | 0.00 / 2.00 | 100 / 3.00 |
| 8 | 0.00 / 2.00 | 0.00 / 2.00 | 0.00 / 0.00 | 0.00 / 2.00 | 0.00 / 2.00 | 100 / 3.00 | 0.00 / 2.00 |
| 9 | 100 / 3.00 | 100 / 3.00 | 0.00 / 0.00 | 100 / 3.00 | 100 / 3.00 | 100 / 3.00 | 100 / 3.00 |
| 10 | 0.00 / 2.00 | 0.00 / 2.00 | 0.00 / 0.00 | 100 / 3.00 | 0.00 / 2.00 | 0.00 / 2.00 | 100 / 3.00 |
| 11 | 100 / 3.00 | 0.00 / 2.00 | 0.00 / 0.00 | 0.00 / 2.00 | 0.00 / 2.00 | 100 / 3.00 | 100 / 3.00 |
| 12 | 100 / 3.00 | 0.00 / 2.00 | 0.00 / 0.00 | 0.00 / 0.00 | 0.00 / 2.00 | 100 / 3.00 | 0.00 / 2.00 |
| 13 | 0.00 / 2.00 | 100 / 3.00 | 0.00 / 0.00 | 100 / 3.00 | 100 / 3.00 | 100 / 3.00 | 100 / 3.00 |
| 14 | 0.00 / 2.00 | 0.00 / 2.00 | 0.00 / 2.00 | 0.00 / 2.00 | 0.00 / 2.00 | 0.00 / 2.00 | 0.00 / 2.00 |
| 15 | 100 / 3.00 | 0.00 / 2.00 | 0.00 / 0.00 | 0.00 / 0.00 | 0.00 / 2.00 | 100 / 3.00 | 100 / 3.00 |

