# OpenReview forum: "Predictive Inverse Dynamics Models are Scalable Learners for Robotic Manipulation"
_ICLR.cc/2025/Conference — ICLR 2025 Oral_

### Official Review · Reviewer_tjHp · 2024-10-21

**Soundness:** 3
**Presentation:** 3
**Contribution:** 3
**Rating:** 6
**Confidence:** 4

**Summary:**

This paper presents a unified approach to combine visual sub-goal prediction and action prediction into a single predictive inverse dynamics model (PIDM).

1. The proposed framework achieves impressive results on LIBERO-LONG and Clavin, two long-horizon manipulation benchmarks.
2. The proposed framework is pretrained on DROID datasets, which shows the potential scalability of PIDM.
3. The authors also evaluate the proposed method with a real robot and conduct ablation studies on the contribution of pretraining, the choices of pre-training, and fine-tuning loss functions.

**Strengths:**

1. This work studied an important problem: improving the data scalability of robot manipulation policies.
2. The idea makes sense. The proposed PIDM combines sub-goal prediction and action prediction. The former, as a vision task, is more generalizable and naturally benefits the latter task. The data efficiency study in Fig.3 is convincing.
3. The results of CALVIN look very impressive.
4. The writing is easy to follow.

**Weaknesses:**

1. The authors did not discuss the limitations of the proposed method at all, while there obviously are. For example, it could be difficult to apply this method in high-frequency control tasks, such as ALOHA. Moreover, the experiments focus on easy manipulation tasks that do not require rich contacts or high precision, such as insertion, screwing, etc.  A thorough discussion of limitations is necessary, especially given the claim that PIDM is a scalable policy.

2. The central idea is not novel. Bringing sub-goal prediction into manipulations has been recently explored by many existing works, such as [1][2][3], especially in the direction of using video generation models for robot tasks.

3. The authors did not explain the network architectures clearly. Some descriptions are in the supplementary, but they are unclear, and there is no figure.

4. When comparing with existing work, the authors only report success rates (or alike), but not network sizes and MACs (number of multiply-additions). It becomes unclear whether the performance gain comes from a larger network or a better method.

[1]: Zero-Shot Robotic Manipulation With Pretrained Image-Editing Diffusion Models. ICLR 2024.

[2]: Learning Universal Policies via Text-Guided Video Generation. NeurIPS 2023.

[3]: Closed-Loop Visuomotor Control with Generative Expectation for Robotic Manipulation. NeurIPS 2024.

**Questions:**

In addition to the weakness section:

1. Could you compare SEER with the existing works that use sub-goal prediction? I am looking for a clear explanation of the essential differences in the method level.

2. In Table 3, how does the finetuning/pretraining work if both the loss functions are not used?

3. How much GPU resources are required during pre-training and fine-tuning?

4. How are sub-goals selected when you prepare the training set?

5. Does this framework work for tasks of more precision or contacts? This is the key issue of using video prediction models in robotics, as these methods sometimes lack the required precision when generating sub-goal videos.  CALVIN has long horizons but very simple manipulation tasks. The LIBERO also seems only have pick and drop, push and pull tasks on relatively large objects.

In general, I appreciate the quality of the idea and result of this work. I rate this paper a borderline accept at this moment, but I have many concerns. Please address them.

---

> ### Author Response · Authors · 2024-11-21
>
> >$\textcolor{brown}{Weakness 1}$ Missing limitations.
>
> We added limitations in **Section 6**:
> "The limitations mainly lie in two aspects. Firstly, we only evaluate 6 downstream tasks. A broader spectrum of high-precision and contact-rich tasks remain to be explored. Secondly, evaluating across different robots is also necessary to test Seer’s cross-embodiments capability."
>
> >$\textcolor{brown}{Weakness2}$  $\textcolor{brown}{and}$  $\textcolor{brown}{Question 1:}$ Comparison with existing sub-goal works.
>
> As shown in table below, our approach outperforms existing sub-goal methods in the CALVIN ABC-D benchmark. The reason is that their sub-goal prediction module and action execution module are separate, and cannot be optimized end-to-end like Seer.
> | Model | Task 1 | Task 2 | Task 3 | Task 4 | Task 5 | Avg. Len.
> | :------: | :------: | :------: | ------: | ------: | ------: | ------:
> | UniPi | 56.0 | 16.0 | 8.0 | 8.0 | 4.0 | 0.92
> | Susie | 87.0 | 69.0 | 49.0 | 38.0 | 26.0 | 2.69
> | Clover | 96.0 | 83.5 | 70.8 | 57.5 | 45.4 | 3.53
> | Ours  | 94.4 | 87.2 | 79.9 | 72.2 | 64.3 | 3.98
>
> >$\textcolor{brown}{Weakness 3:}$ Network architectures.
>
> We have attached the detailed network architecture and descriptions in **Supp. Section A.2**.
> Seer consists of the following modules:
> - image encoder: a MAE pre-trained ViT-Base
> - perceiver resampler: a module to compress the image tokens
> - robot state encoder: MLPs, encoding robot states
> - language encoder: a CLIP ViT-B/32 text encoder
> - transformer backbone: 24 layers of GPT-2 transformer blocks.
> - action decoder: MLPs that decode the latent feature into 7-DOF action.
> - image decoder: a ViT-based transformer followed by a linear layer.
>
> The parameter configuration can be seen in the table below:
> | Module | Hidden size | Number of layers | Number of heads
> | :------: | :------: | :------: | ------:
> | image encoder | 768 | 12 | 12
> | perceiver resampler | 768 | 3 | 8
> | transformer backbone | 384 | 24 | 12
> | image decoder  | 384 | 2 | 16
>
> >$\textcolor{brown}{Weakness 4:}$ Comparison in network sizes and MACs.
>
> We calculate trainable parameters and MACs for baseline methods and ours,  and compare performances across CALVIN ABC-D, LIBERO-LONG, and 4 original real-world tasks. Our method has a larger network but fewer MACs and better performance than GR-1. Our method has the same parameters and MACs as MVP and MPI but surpasses them in simulations and real-world evaluations. Against OpenVLA, our method uses fewer parameters and MACs and performs better in LIBERO-LONG and real tests, confirming its superiority.
>
> + CALVIN ABC-D
> | Method  | Trainable parameters | MACs | Avg. Len.
> | :------: | :------: | :------: | :------:
> | GR-1  | 46.0 M | 358.0 G | 3.06
> | Ours (DROID)  | 65.0 M | 287.5 G | 3.98
>
> + LIBERO-LONG
> | Method | Trainable parameters | MACs | SR
> | :------: | :------: | :------: | :------:
> | MVP  | 65.0 M | 287.5 G |  68.2
> | MPI  | 65.0 M | 287.5 G |  77.3
> | OpenVLA (LoRA, r=32) | 97.6 M | 2.13T | 54.0
> | Ours (DROID) | 65.0 M | 287.5 G | 87.7
>
> + REAL
> | Method  | Trainable parameters | MACs | SR / Score
> | :------: | :------: | :------: | :------:
> | MVP  | 65.0 M | 287.5 G |  55.0 / 29.8
> | MPI  | 65.0 M | 287.5 G |  48.4 / 29.3
> | OpenVLA (Full FT) | 7.54B | 2.13 T | 16.7 / 11.0
> | Ours (DROID) | 65.0 M | 287.5 G | 78.4 / 39.5
>
> >$\textcolor{brown}{Question 2:}$ In Table 3, How does it work without both loss functions?
>
>  The first line omits two loss functions. It uses vanilla behavior cloning, and directly predicts actions from past and present observations.
>
> >$\textcolor{brown}{Question 3:}$ Details about GPU resources.
>
> We used eight RTX 4090 GPUs for pretraining (40 hours for CALVIN ABC-D, 30 hours for LIBERO-LONG) and finetuning (24 and 6 hours, respectively) in simulations. In real-world experiments, the pre-training process costs 32 A100 and 4 days, and fine-tuning process costs 8 A100 and 1 day.
>
> >$\textcolor{brown}{Question 4:}$ Details about sub-goals selection.
>
> The image from the third subsequent frame of each observation serves as the subgoal. For example, at the current time step t, the image at time step t+3 will be regarded as the subgoal.
>
> >$\textcolor{brown}{Question 5:}$ Effectiveness in high-precision and contact-rich tasks?
>
> We added two precision, contact-rich tasks: "Push Button," requiring a robot to press a 2x2cm button and and exceed 3/4 of the scale, and "Insertion," to place a 2x9cm camera into a 2.8x9.5cm groove without collision. As  seen in the table below, DROID pre-training brings obvious improvements, showing Seer's effectiveness in high-precision and contact-rich tasks. More details are in **Supp. Section A.6.2**.
>
> | Method | Press Button | Insertion
> | :------: | :------: | :------:
>  | | SR (\%) / Score | SR (\%)  / Score
> | MVP | 46.7 / 17.0 | 26.7 / 11.0
> | Ours (Scratch) | 40.0 / 13.0 | 40.0 / 16.0
> | Ours (DROID)  | 60.0 / 18.0 | 60.0 / 19.0

---

> > ### Author Response · Authors · 2024-12-01
> >
> > Dear reviewer tjHp:
> >
> > The discussion period is about to end, but we have not received your feedback. We have provided supplementary experiments and responses. We would greatly appreciate the opportunity for further timely discussions and exchanges to address any remaining concerns you might have.
> >
> > Thank you for the time and effort you have invested in providing valuable feedback.

---

### Official Review · Reviewer_dizC · 2024-10-30

**Soundness:** 3
**Presentation:** 3
**Contribution:** 2
**Rating:** 8
**Confidence:** 4

**Summary:**

The paper presents an approach for end2end learning of action and image predictions for imitation learning - termed Predictive Inverse Dynamics Model. The authors present a Transformer architecture that takes in language, observation history, and robot state, and predicts both actions and images in tandem (the action predictions are conditioned on the image predictions). The authors conduct experiments in simulation (LIBERO and CALVIN) and on real-world tasks pretrained on DROID to showcase the benefits of the proposed pretraining approach.

**Strengths:**

Strengths:
* The general problem of learning action/image prediction models or world models is of interest to the community.
* The problem is well-motivated and the literature review does a good job of contextualizing the paper in prior work.
* The paper is strong, well-written and easy to follow.
* The figures are informative and effectively illustrate the benefits of the proposed approach.
* The experiments consider both simulation and real robot evaluation, as well as an ablation study, demonstrating the proposed method's superior performance as compared to the considered baselines.

**Weaknesses:**

* The authors may consider adding the following relevant works to the literature review.
  * Gen2Act: Human Video Generation in Novel Scenarios enables Generalizable Robot Manipulation
  * This&That: Language-Gesture Controlled Video Generation for Robot Planning
  * VideoAgent: Self-Improving Video Generation
  * IGOR: Image-GOal Representations Atomic Control Units for Foundation Models in Embodied AI
* My biggest concern with the paper is the low number of experimental trials in certain settings (e.g., 20) and the lack of indication whether the results are statistically significant. At these lower trial numbers, it would not be unexpected that some of the performance difference is not statistically significant.
* In line 71, it is stated that this approach is the first to optimize vision and action prediction synergistically. However, does the GR-1 paper not do the same high-level goal?
* The limitations of the approach are missing from the paper.
* Line 267 - vanilla typo.

**Questions:**

1. Are the predicted actions at the same frequency as the prediction images? This point was confusing as the actions were referred to as intermediate actions throughout the paper.
2. How many frames of observation history are used as input? How many frames are predicted?
3. Did rolling a chunk of action predictions not work for inference (it seems like only one was rolled out)? Did you end up using a temporal ensemble? How many predictions were in the ensemble?

---

> ### Author Response · Authors · 2024-11-21
>
> > $\textcolor{brown}{Weakness 1:}$ Relevant works.
>
> We added discussions to **Section 2** in the paper:
>
> Another line of work focuses on  visual expectations guiding actions, termed the Predictive Inverse Dynamics Model (PIDM) [1, 2, 3, 4]. Firstly, a video generation model predicts future visual sub-goals and is pre-trained on general visual datasets. Then, an inverse dynamics (goal-conditioned) low-level policy  is trained on downstream robot data to predict intermediary actions.
>
> [1]: Gen2Act: Human Video Generation in Novel Scenarios enables Generalizable Robot Manipulation. Arxiv 2024.
>
> [2]: This&That: Language-Gesture Controlled Video Generation for Robot Planning. Arxiv 2024.
>
> [3]: VideoAgent: Self-Improving Video Generation. Arxiv 2024.
>
> [4]: IGOR: Image-GOal Representations Atomic Control Units for Foundation Models in Embodied AI. Arxiv 2024.
>
>
> >$\textcolor{brown}{Weakness 2:}$ Statistical significance in experiments with low number of trails.
>
> Here we use the t-test to verify the statistical significance in LIBERO-LONG and real-world experiments with low numbers of experimental trials (20 and 15 respectively). We hypothesize that the SOTA baseline, Ours (Scratch) and Ours (DROID)'s success rates satisfy independent normal distribution $N(\mu_1, \sigma_1^2)$, $N(\mu_2, \sigma_2^2)$, $N(\mu_3, \sigma_3^2)$. For all tests, the null hypothesis is rejected when the p value is smaller than the significance level $\alpha=0.01$.
>
> For SOTA baseline and Ours (DROID), we  use the Levene test to verify whether $\sigma_1^2 = \sigma_3^2$. Then we use the t-test. Results:
> | Benchmark | Null Hypothesis | P Value |
> | :------: | :------: | :------: |
> | LIBERO-LONG| $\mu_1 = \mu_3$ | 2.29e-06
> | real-world | $\mu_1 = \mu_3$ | 9.98e-08
>
> In both benchmarks, the p-values are smaller than the significance level $\alpha=0.01$.  Therefore, we reject the null hypothesis, and there exists statistical significance. The sample mean of Ours (DROID) is 87.7 and 71.1, higher than the counterparts (77.3 and 48.9). Therefore, we can regard $\mu_1 < \mu_3$ empirically. This  proves that our method outperforms state-of-the-art baselines.
>
> For Ours (Scratch) and Ours (DROID), we  use the Levene test to verify whether $\sigma_2^2 = \sigma_3^2$. Then we use the t-test. Results:
> | Benchmark | Null Hypothesis | P Value |
> | :------: | :------: | :------: |
> | LIBERO-LONG| $\mu_2 = \mu_3$ | 2.96e-05
> | real-world | $\mu_2 = \mu_3$ | 1.81e-05
>
> In both benchmarks, the p-values are smaller than the significance level $\alpha=0.01$.  Therefore, we reject the null hypothesis, and there exists statistical significance. The sample mean of Ours (DROID) is 87.7 and 71.1, higher than the counterparts (78.7 and 53.3) in LIBERO-LONG and real-world experiments. Therefore, we can regard $\mu_2 < \mu_3$ empirically.  This  proves the effectiveness of pre-training.
>
>
> >$\textcolor{brown}{Weakness 3:}$ Does GR-1 optimize vision and action prediction synergistically.
>
> Seer is pre-trained and fine-tuned on robot datasets, enabling optimization vision and action prediction during both phases.  In comparison, GR-1 model is pre-trained on the Ego-4d dataset and only supervised via vision loss during pre-training.
>
> Furthermore, Seer optimizes vision and action loss in a synergistic way, where action predictions are dependent on vision predictions, optimizing in an end-to-end manner. In comparison, vision prediction in GR-1 only serves as an auxiliary loss and does not serve as a direct condition for action execution.
>
>
>
> >$\textcolor{brown}{Weakness 4:}$ Missing limitations.
>
> We have added limitations to **Section 6** in the paper:
>
> "The limitations mainly lie in two aspects. Firstly, we only evaluate 6 downstream tasks. A broader spectrum of high-precision and contact-rich tasks remain to be explored. Secondly, evaluating across different robots is also necessary to test Seer’s cross-embodiments capability."
>
> >$\textcolor{brown}{Weakness 5:}$ Typo.
>
> The typo has been corrected.
>
>   >$\textcolor{brown}{Question 1:}$ Prediction frequency and definition of intermediate actions.
>
> The predicted frequency of "action" is three times that of "image". The term "intermediate action" refers to the action required to transition from the current time step to the future time step. Concretely, at each time step t, the image at t + 3 will be forecasted, and actions at t, t+1, t+2 will be predicted.
>
> >$\textcolor{brown}{Question 2:}$ Numbers of observation history and predicted frames.
>
> The history length for CALVIN is 10, and that of LIBERO and real world experiments is 7. 3 future frames are predicted for all  three benchmarks.
>
> >$\textcolor{brown}{Question 3:}$ Action chunking and temporal ensembling.
>
> During Inference, we implement  with a action chunk size of 3, and we apply the temporal ensembling across 3 predictions.

---

> > ### Comment · Reviewer_dizC · 2024-11-25
> > **Response to Rebuttal**
> >
> > Thank you to the authors for the thoughtful responses! The clarifications were helpful. I will keep my score as it is already high. I do encourage the authors to include limitations that are inherent to the method rather than simply require further experimentation.

---

### Official Review · Reviewer_f4m6 · 2024-10-30

**Soundness:** 3
**Presentation:** 3
**Contribution:** 3
**Rating:** 8
**Confidence:** 4

**Summary:**

This paper proposes an end-to-end Predictive Inverse Dynamics Model (PIDM) paradigm, to integrate vision and robot actions in a closed loop for a scalable policy for robotic manipulation. They introduce the Seer model, designed to leverage visual, temporal, and action features for large-scale pretraining, effectively combining the advantages of vision and action in policy formulation. The model is structured around two key components: Conditional Visual Foresight and Inverse Dynamics Prediction.

In Conditional Visual Foresight, Seer generates future RGB image predictions conditioned on specified goals (such as language instructions or robot states) and recent observations (previous RGB images and robot states), allowing the robot to anticipate and guide its actions. The Inverse Dynamics Prediction module then takes two consecutive observations—the current state and the visually predicted future state—to infer the required action sequence to achieve the anticipated outcome. This allows Seer to dynamically adjust its actions based on updated visual predictions, maintaining adaptability and responsiveness.

They performed extensive experiments in simulations and the real world, demonstrating the generalization capabilities and robust performance of the proposed method, underscoring its potential in advancing scalable, closed-loop policy solutions for robotic manipulation.

**Strengths:**

- Effective Pre-training Objectives: The paper emphasizes the importance of pre-training the entire policy rather than just visual components, leading to significant performance improvements. The conditional visual foresight objective encourages the model to learn from the future, while the inverse dynamics prediction objective helps it understand the relationship between actions and their visual consequences.
- End-to-End Training: Both the visual prediction and inverse dynamics modules are trained together, enabling the model to effectively utilize visual, temporal, and action information from large datasets. Seer uses Transformers to process visual states and actions, making it scalable.
- Unidirectional Attention: The model uses a unidirectional attention mask in the Transformer encoder, allowing the action prediction module to integrate past and future information for better decision-making.
- Extensive Results: The paper provides extensive experimental results on both simulated and real-world benchmarks. Seer consistently outperforms state-of-the-art baselines on tasks involving long-horizon planning, generalization to unseen scenes, and robustness to disturbances.

**Weaknesses:**

- Generalization across embodiment: While this paper proposes a generalizable ene-to-end pre-training paradigm for robotic manipulation policy, the proposed Seer model is pre-trained on the DROID [1] dataset, which only involves the Franka Panda robot. Training on the dataset across different embodiments such as Open-X [2] might improve generalization capability.

- In the real-world evaluation, they selected dataset DROID, which contains demonstrations of Franka robots executing various tasks in diverse scenes. It is likely to overfit the dataset for policies for different tasks.

[1] Khazatsky, Alexander, et al. "Droid: A large-scale in-the-wild robot manipulation dataset." arXiv preprint arXiv:2403.12945 (2024).

[2] O.-X. E. Collaboration, A. Padalkar, A. Pooley, A. Jain, A. Bewley, A. Herzog, A. Irpan, A. Khazatsky, A. Rai, A. Singh, et al., “Open x-embodiment: Robotic learning datasets and rt-x models,” arXiv preprint arXiv:2310.08864, 2023.

**Questions:**

- Generalization Across Embodiment: The model is pre-trained on the DROID dataset, limited to the Franka Panda robot. How does this impact its generalization to different robot types? Would you please discuss the possibility of training on a more diverse dataset, like Open-X, to improve adaptability?

- Real-World Evaluation: The real-world evaluation uses the DROID dataset, with only Franka robots in varied tasks. Is there a risk of overfitting, potentially limiting performance in broader tasks? Would you please discuss task generalization?

- Regarding the OpenVLA baseline, which specific checkpoint version was used? OpenVLA has multiple versions fine-tuned on LEBERO, and their performance varies across different tasks.

- Typo: Line 067 using Inverse Dynamics Models (IDM) conditioned on the robots’. There is an extra "'".

---

> ### Author Response · Authors · 2024-11-21
>
> >$\textcolor{brown}{Weakness 1}$  $\textcolor{brown}{and}$ $\textcolor{brown}{Question 1:}$ Generalization across embodiments.
>
> Here we add the real-world experiment, pre-trained on OXE and fine-tuned on self-collected data. We refer the subset mix-up recipe in Octo [1], remove all the subset that includes franka robots, filter subsets with odd action labels,  and save the rest subsets as OXE pre-training dataset. We also add 2 challenging tasks, Press Button and Insertion, that require high precision and rich contact. The results are listed as follows:
> | Method | Flip White Bowl | Stack Cups | Wipe Board | Pick, Place, Close | Press Button | Insertion | Avg.
> | :------: | :------: | :------: | :------: | :------: | :------: | :------: | :------:
> | | SR (\%) / Score | SR (\%)  / Score | SR (\%)  / Score | SR (\%)  / Score | SR (\%)  / Score | SR (\%)  / Score | SR (\%) $\uparrow$   / Score $\uparrow$
> | MVP  | 80.0 / 24.0 | 26.7 / 26.0 | 53.3 / 38.0 | 60.0 / 31.0 | 46.7 / 17.0 | 26.7 / 11.0 | 48.9 / 24.5
> | Ours (Scratch) | 60.0 / 19.0 | 46.7 / 35.0 | 60.0 / 37.0 | 73.3 / 40.0 | 40.0 / 13.0 | 40.0 / 16.0 | 53.3 / 26.7
> | Ours (OXE) | 73.3 / 22.0 | 40.0 / 30.0 | 66.7 / 40.0 | 80.0 / 41.0 | 33.3 / 10.0 | 46.7 / 18.0 | 56.7 / 26.8
> | Ours (DROID) | 86.7 / 26.0 | 60.0 / 42.0 | 73.3 / 41.0 | 86.7 / 42.0 | 60.0 / 18.0 | 60.0 / 19.0 | 71.1 / 31.3
>
> As can be seen in the table, pre-training on the OXE dataset only provides marginal improvements in most tasks. Moreover, in some high-precision tasks, the OXE pre-trained version even brings negative effects. We attribute marginal improvements in general manipulation tasks to the diversity of objects, tasks, scenes, and language instructions in OXE.  For the slight decrease in some high-precision tasks, we suspect two reasons:
> - For high-precision tasks, images from the eye-on-hand camera (wrist view) are quite important, since this camera captures a lot of local and detailed information for policy. However, in most OXE subsets, only images from third-person view with diverse camera poses are provided, with most wrist view images missing.
> - Admittedly, the gap in cross-embodiments and  cross-action-controller types exists in our settings. High-precision tasks require a concentrated and precise action distribution. However, pre-training on OXE may offer a distant action prior due to the aforementioned physics gap.
>
> [1]: Octo: An open-source generalist robot policy. Arxiv 2024.
>
>
>
> >$\textcolor{brown}{Weakness 2}$  $\textcolor{brown}{and}$ $\textcolor{brown}{Question 2:}$ Risk of overfitting DROID dataset and discussion about task generalization.
>
> The DROID  dataset is diverse, containing 76k episodes, 564 scenes, 52 buildings and 86 tasks or verbs. Therefore, it is difficult for a model to overfit a specific task. We believe it is our method's ability of distilling action and vision priors from DROID that brings obvious improvements in downstream tasks. Besides, in our six (4 original + 2 newly added) downstream tasks, only limited objects (e.g. bowls and cups) and tasks (e.g. simple pick and place) may be seen in DROID, while most objects (e.g. small delicate items, dustpan, brush, insertion-related objects) and challenging tasks (e.g. precisely stack, insert, press, sweep) are rare in DROID. In this way, our method could remain a satisfying result, demonstrating great objects and tasks generalization.
>
>
> >$\textcolor{brown}{Question 3:}$ OpenVLA's version on LIBERO.
>
> We utilize the OpenVLA-7B checkpoint, fine-tuned via LoRA (r=32) on LIBERO-LONG, and adhere to the official instructions for launching LIBERO evaluations. The checkpoint is available at https://huggingface.co/openvla/openvla-7b-finetuned-libero-10.
>
> >$\textcolor{brown}{Question 4:}$ Typo.
>
> Thanks for your sincere advice. The typo has been corrected.

---

> > ### Comment · Reviewer_f4m6 · 2024-11-25
> >
> > I thank the authors for their thoughtful rebuttal and the experiments provided for the rebuttal. I see this as a strong paper with no major concerns from my perspective. The insight to incorporate inverse dynamics is great and the experiment results are solid. I am pleased to raise my rating and recommend it for acceptance.

---

> > > ### Author Response · Authors · 2024-11-27
> > >
> > > Thank you for considering our responses and recommending acceptance. We appreciate the time and effort you have dedicated to reviewing our work.

---

### Official Review · Reviewer_zoME · 2024-10-31

**Soundness:** 3
**Presentation:** 3
**Contribution:** 3
**Rating:** 8
**Confidence:** 4

**Summary:**

This paper introduces an end-to-end framework, Predictive Inverse Dynamics Models (PIDM), which predicts actions using inverse dynamics models conditioned on the robot’s forecasted visual states. By integrating vision and action in a closed-loop system, the end-to-end PIDM serves as a scalable and effective action learner. The approach demonstrates improved performance over state-of-the-art methods, both in simulations and on a real robot.

**Strengths:**

1. Thorough ablation studies and experiments that demonstrate the effectiveness of jointly predicting forward and inverse dynamics as a pre-training task for robot learning
2. Evaluation in both simulation and in real on the effectiveness of the method
3. Shows scalability of the proposed method by pre-training on DROID.

**Weaknesses:**

1. Learning inverse dynamics models from visual inputs has been explored in the past (i.e. [1,2]). It would be good to discuss these papers in the context of this paper.
2. The paper shows scalability in the direction of pre-training and finetuning data. To fully demonstrate scalability, it would be good to demonstrate the scalability in the model capacity axis as well.
3. The current formulation of the model does not seem to take account of the history (past observations). This makes it challenging to extend to more complex environments where stronger task planning is needed.

[1] Agrawal, Pulkit, Ashvin V. Nair, Pieter Abbeel, Jitendra Malik, and Sergey Levine. "Learning to poke by poking: Experiential learning of intuitive physics." Advances in neural information processing systems 29 (2016).[1] Agrawal, Pulkit, Ashvin V. Nair, Pieter Abbeel, Jitendra Malik, and Sergey Levine. "Learning to poke by poking: Experiential learning of intuitive physics." Advances in neural information processing systems 29 (2016).

[2] Brandfonbrener, David, Ofir Nachum, and Joan Bruna. "Inverse dynamics pretraining learns good representations for multitask imitation." Advances in Neural Information Processing Systems 36 (2024).

**Questions:**

1. The joint prediction of inverse dynamics (actions) and forward dynamics (next observation) reminds me of RPT [1], where it performs random masking of action and latent representations of observations and pre-trained via mask reconstruction. The key difference between this approach and RPT is that RPT doesn’t predict exact forward dynamics due to the random mask pattern, so it can learn shortcuts by copying from other future frames. Is there some way to evaluate the importance of predicting forward dynamics? Specifically, for table 3(b): is it possible to pre-train with only inverse dynamics?
2. For finetuning the model, are all weights considered, or is LoRA applied to the model? It would be interesting to know how this method's performance scales with the amount of parameters used for fine-tuning.

[1] Radosavovic, Ilija, Baifeng Shi, Letian Fu, Ken Goldberg, Trevor Darrell, and Jitendra Malik. "Robot learning with sensorimotor pre-training." In Conference on Robot Learning, pp. 683-693. PMLR, 2023.

[2] Hu, Edward J., Yelong Shen, Phillip Wallis, Zeyuan Allen-Zhu, Yuanzhi Li, Shean Wang, Lu Wang, and Weizhu Chen. "Lora: Low-rank adaptation of large language models." arXiv preprint arXiv:2106.09685 (2021).

---

> ### Author Response · Authors · 2024-11-21
>
> >$\textcolor{brown}{Weakness 1:}$ Discuss works using inverse dynamics models in the past.
>
> Thanks for your sincere advice. We have discussed in Section 2 as follows:
>
> Some studies [1, 2] integrate current and goal information to extract effective features or serve as an auxiliary objective. Subsequently, a standard behavior cloning approach is applied during downstream task implementations.
>
> >$\textcolor{brown}{Weakness 2:}$ Demonstrate the scalability in the model capacity axis.
>
> We perform validation across four model sizes using the CALVIN ABC-D benchmark, with each model trained from scratch. The results for the Avg.Len. metric are presented in the table below. It is observed that performances improve with the increase in model size.
> | | | | | |
> | :-----: | :-----: | :-----: | :-----: | :-----: |
> | Model size | 44M | 65M | 107M | 316M |
> | w/o pretrain | 3.32 | 3.64 | 3.81 | 3.85 |
> | w/ pretrain | 3.55 | 3.98 | 4.12 | 4.29 |
>
>
> >$\textcolor{brown}{Weakness 3:}$ Whether taking account of history?
>
> The original model has already taken historical inputs into account. We apply varying history lengths across different benchmarks: for CALVIN, the history length is 10, and for LIBERO and real-world experiments, it is 7. The details are outlined in Table 5 in the Appendix.
>
>
> >$\textcolor{brown}{Question 1:}$ Pre-train with inverse dynamics only.
>
>   Here we add an additional ablation experiment in Table 3(b). During pre-training, only the inverse dynamics loss function is used, where actions are inferred from history and **ground truth** future information. We remain the original fine-tuning objectives. As can be seen below, being pre-trained via only inverse dynamics brings improvements, which is also validated in [1]. However, being pre-trained via both inverse dynamics and forward dynamics yields better improvements, demonstrating our method's superiority.
>
> | $\mathcal{L}_{\text{fore}}$ | $\mathcal{L}_{\text{inv}}$ | Task 1 | Task 2 | Task 3 | Task 4 | Task 5 | Avg. Len.
> | :------: | :------: | :------: | ------: | :------: | :------: | :------: | :------:
> | $\times$ | $\times$ | 93.0 | 82.4 | 72.3 | 62.6 | 53.3 | 3.64
> | $\checkmark$ | $\times$ | 92.3 | 83.0 | 74.2 | 65.9 | 57.5 | 3.73
> | $\times$ | $\checkmark$ | 93.8 | 84.2 | 74.3 | 65.9 | 57.0 | 3.75
> | $\checkmark$ | $\checkmark$ | 94.4 | 87.2 | 79.9 | 72.2 | 64.3 | 3.98
>
> [1] Inverse dynamics pretraining learns good representations for multitask imitation. Neurips 2024.
>
> >$\textcolor{brown}{Question 2:}$ How to fine-tune the model?
>
> We finetune all parameters except those of the CLIP text encoder and MAE-pretrained VIT-Base vision encoder. We do not apply LoRA , since finetuning all trainable parameters (65M) is affordable and effective.

---

### Author Response · Authors · 2024-11-21

We thank all the Reviewers for their detailed and helpful comments on our work. We appreciate the reviewers for acknowledging our strengths and contributions. During the rebuttal phase, we have made diligent efforts to address the concerns raised by the reviewers. Specifically:
- We added a scalability evaluation regarding the model size.
- We added an experiment where Seer is pre-trained with only inverse dynamics objective.
- We performed real-world experiments where Seer is pre-trained on a diverse and cross-embodiments dataset, Open X-Embodiment.
- We added two real-world experiments that require high-precision and contact-rich manipulation (**Supp. Section A.6.2, Figure 8, Table 7, LIne 927-967**).
- We validated the statistical significance in evaluations with low trail numbers.
- We compared our method with previous sub-goal methods in the simulation.
- We included a detailed network architecture description (**Supp Section A.2, Figure 6, Table 6, Line 810 - 831**).
- We added more literature reviews (**Paper Section 2, Line 95 - 97 and Line 131 - 136**), limitations (**Paper Section 6, Line 537 - 539**), and fixed all the mentioned typos in the paper.

An updated version of the paper and supplementary materials is uploaded and major changes are highlighted in blue.

---

### Meta-Review · Area_Chair_1jLz · 2024-12-25

**Metareview:**

The paper introduces Seer, a Predictive Inverse Dynamics Model (PIDM) that integrates vision and action learning in an end-to-end framework using Transformers, achieving state-of-the-art performance and strong generalization across simulation and real-world robotic manipulation tasks.

The reviewers unanimously recognized the paper's contributions, highlighting its (1) comprehensive ablation studies, (2) evaluations in both simulation and real-world settings, (3) demonstration of scalability, and (4) clear and well-structured presentation.

During the Author-Reviewer Discussion phase, the authors provided detailed responses that successfully addressed many of the reviewers' concerns, leading to score increases from some reviewers. With all reviewers in unanimous agreement to accept the paper, the AC recommends acceptance while encouraging the authors to carefully address both pre- and post-rebuttal comments to further strengthen the final version.

**Additional Comments On Reviewer Discussion:**

Since the reviewers were in unanimous agreement to accept this paper, no significant discussion took place during the Reviewer Discussion phase.

---

### Decision · Program_Chairs · 2025-01-22

Accept (Oral)